# Stochastic optimal feedforward-feedback control determines timing and variability of arm movements with or without vision

**Bastien Berret**[1,2,3]*, **Adrien Conessa**[1,2], **Nicolas Schweighofer**[4], **Etienne Burdet**[4]

**1** Université Paris-Saclay CIAMS, Orsay, France, **2** CIAMS, Université d'Orléans, Orléans, France, **3** Institut Universitaire de France, Paris, France, **4** University of Southern California, Los Angeles, California, United States of America, **5** Imperial College of Science, Technology and Medicine, London, United Kingdom

* bastien.berret@universite-paris-saclay.fr

**Data Availability Statement:** All relevant experimental data are within the manuscript and its Supporting information files.

## Abstract

Human movements with or without vision exhibit timing (i.e. speed and duration) and variability characteristics which are not well captured by existing computational models. Here, we introduce a stochastic optimal feedforward-feedback control (SFFC) model that can predict the nominal timing and trial-by-trial variability of self-paced arm reaching movements carried out with or without online visual feedback of the hand. In SFFC, movement timing results from the minimization of the intrinsic factors of effort and variance due to constant and signal-dependent motor noise, and movement variability depends on the integration of visual feedback. Reaching arm movements data are used to examine the effect of online vision on movement timing and variability, and test the model. This modelling suggests that the central nervous system predicts the effects of sensorimotor noise to generate an optimal feedforward motor command, and triggers optimal feedback corrections to task-related errors based on the available limb state estimate.

## Author summary

Stochastic optimal feedback control, which has been extensively used to model human motor control in the last two decades, proposes to compute an optimal motor command online based on an estimation of the current system state using sensory feedback. However, this modelling approach underestimates the role of motor plans to generate appropriate feedforward motor command before the movement starts, which is emphasized in conditions with large uncertainty about current limb state estimates such as when visual feedback is lacking. Here we propose a model combining stochastic feedforward and feedback control to address this issue. The new stochastic feedforward-feedback (SFFC) model considers effort and variance minimization as well as the effects of motor and sensory noise both on planning and execution of arm movements. By combining the feedforward and feedback aspects of stochastically optimal control in an elegant way, SFFC can predict the timing and variability of movements carried out with or without visual

**Funding:** This work was in part funded by the EC under grants H2020 PH-CODING (FETOPEN 829186), INTUITIVE (ITN 861166), REHYB (ICT 871767) [EB], by NIH NINDS 1R56NS100528, NIH NINDS R21NS120274 grants [NS], and by the French National Agency for Research (grant ANR-19-CE33-0009) [BB]. The funders had no role in study design, data collection and analysis, decision to publish, or preparation of the manuscript.

**Competing interests:** The authors have declared that no competing interests exist.

feedback, while previous models would fail in one or another aspect, or have to use ad hoc fixes.

## Introduction

Spatial and temporal regularities in human motion suggest that the neural control of movement involves a planning stage [1, 2]. Evidence for motor planning has been provided in behavioural experiments [3, 4] and through the observation of neural processes prior to movement generation [5–7]. Among the planned aspects of movement, the timing (i.e. speed and duration) and trial-by-trial variability are important determinants of successful actions [8]. However, the principles according to which the central nervous system (CNS) may determine these critical features is not well explained by existing models.

The currently dominating theory of motor control, stochastic optimal (feedback) control (SOC) [9–12], can explain the coordination of the degrees-of-freedom of the sensorimotor system, the structure of trial-by-trial variability or the reactive behavior to external perturbations [13–16]. However, SOC does not account well for the timing of self-paced reaching movements. As with deterministic optimal control (DOC) models (e.g. [17, 18]) the costs minimised in SOC models, effort and error, decrease monotonically with increasing movement duration, thereby predicting infinitely slow visually-guided movements. Fig 1A illustrates this issue for the SOC model of [19] where both motor and observation noise are considered. Interestingly, when SOC is used to model movements without vision by increasing observation noise to reflect a degraded hand state estimate, a finite optimal duration can be obtained because endpoint variance now increases with duration. Therefore, SOC with large enough observation noise may determine the timing and variability of movements without vision (Fig 1B), but the same principle cannot be used directly for visually-guided movements.

Several ad hoc solutions have been proposed to circumvent this issue. In several models considering sensorimotor noise, duration was selected as the minimum time to match a desired endpoint variance related to target's width, based on the speed-accuracy trade-off underlying visually-guided movements [8, 20–22]. Alternatively, a number of studies have assumed a "cost of time" (reflecting neuroeconomical processes related to decision-making and explicitly penalizing duration) to explain the preferred timing of movement [23–30].

However, the preferred movement timing may be predicted without requiring an ad hoc solution. In particular, the results of [31] suggest that the motor noise, with its constant and signal dependent components, is a relevant factor to determine this characteristic of motion planning. Specifically, the preferred duration of movements performed without vision was found to be longer than the minimum variance duration, thereby suggesting that movement timing was determined from a neuromechanical principle based on a trade-off between effort and variance in the presence of signal-dependent and constant motor noise. Such optimality principle could explain the stereotyped durations and trajectories of saccades [32], but its relevance for arm reaching has not been tested in particular for visually-guided movements.

Because SOC cannot be used for this purpose (see Fig 1A), here we develop a new computational model to predict the timing and variability of arm pointing movements carried out with complete or degraded sensory feedback (e.g. when vision of the hand is prevented) from neuromechanical factors only. This *stochastic feedforward-feedback control* (SFFC) model assumes that the motor command comprises a feedforward and a feedback components. The feedforward component is computed using the stochastic optimal open-loop control (SOOC) framework, which was initially developed to account for the planning of mechanical impedance via

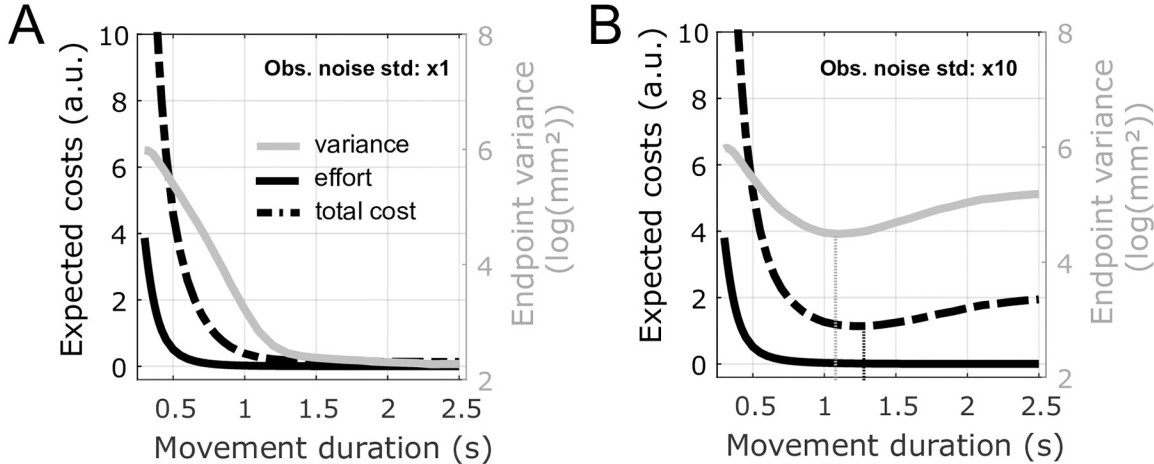

**Fig 1. Expected costs and endpoint variance for the SOC model of [19] for simulated movements with (A) and without (B) vision.**
A 10-cm long reaching movement of a point mass model of an arm is simulated. These simulations rely on the extended linear-quadratic-Gaussian framework considering multiplicative (signal-dependent) and additive (constant) motor noise as well as additive observation noise. A. Simulations with a standard observation noise corresponding to a visually-guided movement, as proposed in the original model of [19]. The expected costs for different movement durations were estimated using the Monte Carlo method (100,000 samples). This model fails to predict a finite movement duration because the optimal expected effort and total cost (sum of effort and terminal error costs) monotonically decrease with duration and plateau. The positional endpoint variance (gray trace) can also be seen to decrease and plateau to a value which mainly corresponds to that of visually-guided movements. B. Simulations with a large observation noise (noise in A multiplied by 10), corresponding to movements without vision. In this case, an optimal duration can be determined as the minimum of the total cost (indicated by a black vertical dotted line).

muscle co-contraction [33, 34]. This feedforward command yields an expectation about a timed trajectory. The feedback component is then computed using the linear SOC framework from a local approximation of the task dynamics, which allows triggering motor corrections in reaction to deviations from the goal, based on an estimation of the system's state from the available sensory information and internal predictions. As a result, the proposed model merges the main precepts of influential models highlighting either the role of feedforward-only or feedback-only control [8–10, 13, 17–19].

Predictions from the SFFC model are first tested by simulating arm reaching movements carried out without visual feedback and comparing the results with the available experimental results of [31, 35], and [36]. Second, an experiment was conducted to analyse the timing and variability of movements performed with and without online visual feedback of the hand. The SFFC predictions for movements in these two conditions are then compared to these new experimental data.

## Results

### Stochastic feedforward-feedback control model

In the proposed model, the actual motor command is made of a *feedforward component* (i.e. determined prior to movement execution) and a *feedback component* (i.e. determined throughout movement execution based on an estimation of the current state) to correct task-related errors as illustrated in Fig 2. This is a classical approach in optimal control theory (e.g. see [37, 38]). However, feedforward control is usually associated with deterministic systems and feedback control with stochastic systems. In the approach presented here, the feedforward command is optimized for the system's stochasticity (i.e. presence of both signal-dependent and constant motor noise) as in [8] and [33].

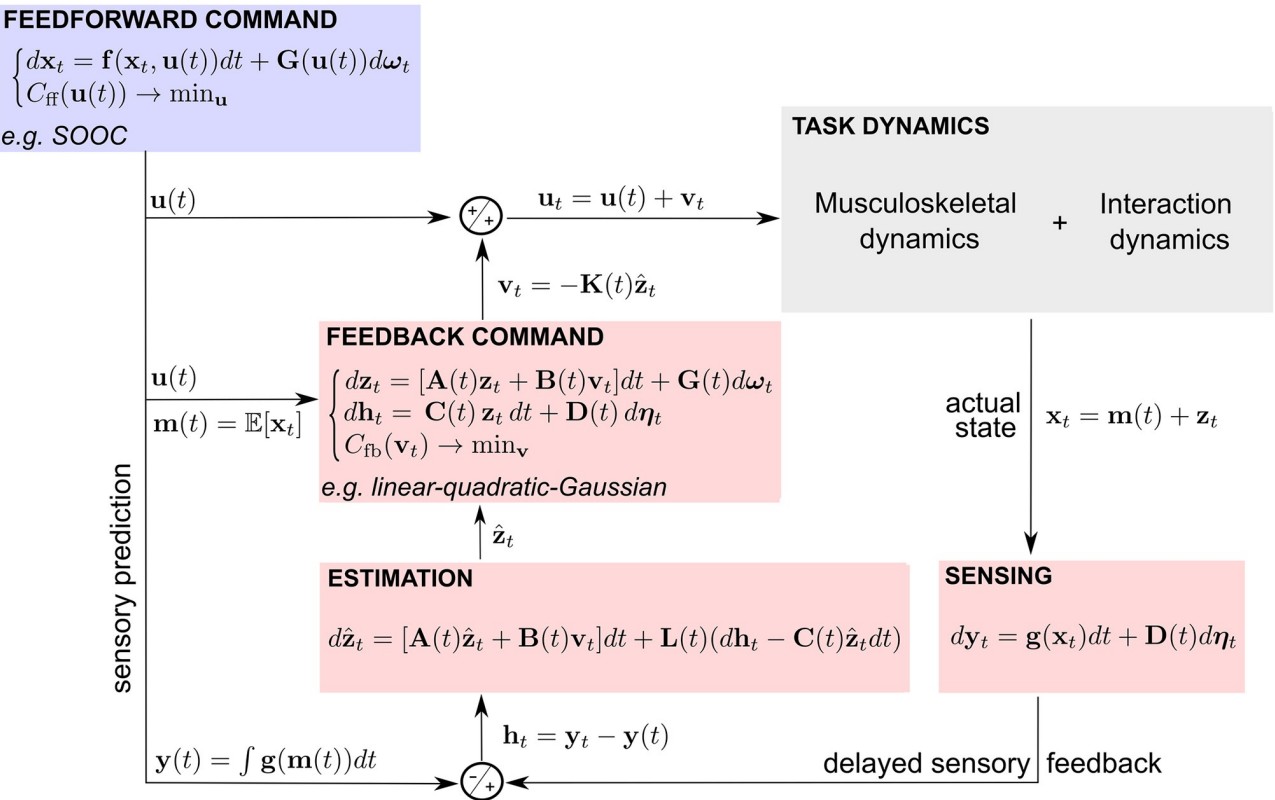

**Fig 2. Scheme of stochastic optimal feedforward-feedback control (SFFC).** A feedforward command $\mathbf{u}(t)$ is formed by the CNS based on prior knowledge about the task dynamics represented by $\mathbf{f}$ and $\mathbf{G}$. A representation of the associated expected system's trajectory $\mathbf{m}(t) = \mathbb{E}[\mathbf{x}_t]$ is also established, which allows building a local approximation of the task dynamics and working in terms of state/control deviations ($\mathbf{z}_t$ and $\mathbf{v}_t$ respectively) during movement execution. This is done by setting the matrices $\mathbf{A}(t) = \frac{\partial \mathbf{f}}{\partial \mathbf{x}}(\mathbf{m}(t), \mathbf{u}(t))$ and $\mathbf{B}(t) = \frac{\partial \mathbf{f}}{\partial \mathbf{u}}(\mathbf{m}(t), \mathbf{u}(t))$. An estimate of the current state deviation $\hat{\mathbf{z}}_t$ is computed from multisensory information $\mathbf{y}_t$. This allows triggering a feedback command online to correct task-relevant errors caused by unexpected internal and external perturbations due e.g. to motor noise or external forces. In this scheme, the actual motor command $\mathbf{u}(t) + \mathbf{v}_t$ is the sum of the feedforward and feedback commands. The matrices $\mathbf{L}(t)$ and $\mathbf{K}(t)$ denote the optimal filter and feedback gains respectively, $\mathbf{g}$ is the output function and $\mathbf{C}(t) = \frac{\partial \mathbf{g}}{\partial \mathbf{x}}(\mathbf{m}(t))$ in the local approximation. The random processes $\boldsymbol{\omega}_t$ and $\boldsymbol{\eta}_t$ are implemented here as Brownian. $\mathbf{D}(t)$ is an observation noise matrix, the magnitude of which can be increased to simulate the absence of vision. The vector $\mathbf{h}_t$ denotes the deviation from the sensory prediction. The terms $C_{\text{ff}}$ and $C_{\text{fb}}$ refer to the cost functions that determine the optimal signature of the feedforward and feedback commands. The definition and meaning of all the variables are given in the Results section.

As we shall see, considering a feedforward command for a stochastic plant allows predicting an optimal movement duration because this command considers the effects of additive noise in the temporal evolution of the state covariance. The actual movement timing and variability may however be affected by sensory-based motor corrections issued online to handle unexpected perturbations and critical deviations from the task's goal. This is the role of the high-level feedback command. In SFFC, the motor plan is thus primarily composed of a feedforward motor command (i.e. an optimal open-loop control) and an expectation about the upcoming state trajectory. It is complemented by a locally-optimal feedback gain that combines with a limb state estimate throughout movement execution to determine a task-relevant corrective motor command. This estimate is based on both internal predictions and relevant sensory information (e.g. proprioception or vision). We describe below how the feedforward and feedback components of the model are computed.

**Determining the feedforward motor command via nonlinear stochastic optimal open-loop control.** Here we consider a minimum effort-variance model of motor planning with

additive and multiplicative motor noise to determine the feedforward motor command. The expectation and covariance of a nominal state trajectory can be obtained from this sub-problem.

Let us consider a general rigid body dynamics with $n$ degrees of freedom such as to model human arm movements:

$$\boldsymbol{\tau} = \mathcal{M}(\mathbf{q})\,\ddot{\mathbf{q}} + \mathcal{C}(\mathbf{q}, \dot{\mathbf{q}})\,\dot{\mathbf{q}} + \mathcal{B}\,\dot{\mathbf{q}} + \mathcal{G}(\mathbf{q})\,, \tag{1}$$

where $\mathbf{q} \in \mathbb{R}^n$ is the joint coordinates vector, $\boldsymbol{\tau} \in \mathbb{R}^n$ the net joint torque vector produced by muscles, $\mathcal{M}$ the inertia matrix, $\mathcal{C}\dot{\mathbf{q}}$ the Coriolis/centripetal, $\mathcal{B}\dot{\mathbf{q}}$ the viscosity, and $\mathcal{G}$ the gravity terms. This dynamical system is nonlinear due to the mechanical coupling between the different body segments and gravity. Let us assume that the torque change is the control variable as in [18]:

$$\mathbf{u} = \dot{\boldsymbol{\tau}}\,. \tag{2}$$

In a standard SOC model the motor command would be a stochastic variable $\mathbf{u}_t$ that depends on the random fluctuations arising from motor and measurement noise as well as from any environmental perturbations. As stressed before and in Fig 2, here we rather assume that motor planning primarily builds a feedforward motor command. This enables the feedforward component of the motor command to consider all the internal and external dynamic effects that can be learnt (including the consequences of noise, the sensory delays, the instability due to the interaction with the environment etc.) during the planning stage. To derive such a feedforward motor command, we restrict the control to be open-loop (denoted by $\mathbf{u}(t)$ to stress its deterministic nature) while retaining the stochastic aspect of the system's dynamics.

To this aim, let us assume that the arm movements are affected by multiplicative motor noise (i.e. with signal-dependent variance) and additive motor noise (i.e. with constant variance), modeled as a $M$-dimensional standard Brownian motion, $\boldsymbol{\omega}_t$. The corresponding stochastic dynamics of the arm can be described by

$$d\mathbf{x}_t = \mathbf{f}(\mathbf{x}_t, \mathbf{u}(t))dt + \mathbf{G}(\mathbf{u}(t))\,d\boldsymbol{\omega}_t\,, \quad \mathbf{x}_t = \begin{bmatrix} \mathbf{q}_t \\ \dot{\mathbf{q}}_t \\ \boldsymbol{\tau}_t \end{bmatrix}, \tag{3}$$

where $\mathbf{x}_t$ is the stochastic state vector and $\mathbf{f} \in \mathbb{R}^N$ and $\mathbf{G} \in \mathbb{R}^{N \times M}$ are respectively the drift and diffusion terms (here $N = 3n$). The matrix $\mathbf{G}$ includes both the constant and the signal-dependent noise terms.

For the reaching task under consideration, the control objective is to move the arm from an initial position $\mathbf{x}_0$ to a given target in time $T$ with minimum effort and minimum variance, that is, by minimizing an expected cost of the form

$$C_{\mathrm{ff}}(\mathbf{u}(t)) = \mathbb{E}\left[r\,\phi(\mathbf{m}(T), \mathbf{x}_T) + \int_0^T l(\mathbf{m}(t), \mathbf{u}(t))\,dt\right], \tag{4}$$

where $\phi$ is a quadratic function penalizing the final state of the process (typically related to its covariance here), and $l$ is a cost depending on $\mathbf{m}(t) = \mathbb{E}[\mathbf{x}_t]$ and the open-loop control $\mathbf{u}(t)$. The cost $l$ can be thought as a measure of effort but it can also include terms like trajectory smoothness. The parameter $r$ is a weighting factor to trade-off the variance and effort/trajectory costs.

This stochastic optimal open-loop control problem can be solved using approximate solutions based on stochastic linearization techniques (see [33]).

Let us denote the covariance of the process $\mathbf{x}_t$ by

$$\mathbf{P}(t) = \mathbb{E}[\mathbf{z}_t \mathbf{z}_t'], \quad \mathbf{z}_t = \mathbf{x}_t - \mathbf{m}(t).\tag{5}$$

It can be shown (e.g. [39], Chap. 12) that propagation of the mean $\mathbf{m}(t)$ and covariance $\mathbf{P}(t)$ can be approximated using a second order Taylor's expansion for $\mathbf{f}$ by the following ordinary differential equations:

$$\dot{\mathbf{m}}(t) = \mathbf{f}(\mathbf{m}(t), \mathbf{u}(t)) + \frac{1}{2}\frac{\partial^2 \mathbf{f}}{\partial \mathbf{x}^2}(\mathbf{m}(t), \mathbf{u}(t)) \bullet \mathbf{P}(t),$$

$$\dot{\mathbf{P}}(t) = \frac{\partial \mathbf{f}}{\partial \mathbf{x}}(\mathbf{m}(t), \mathbf{u}(t))\mathbf{P}(t) + \mathbf{P}(t)\frac{\partial \mathbf{f}}{\partial \mathbf{x}}(\mathbf{m}(t), \mathbf{u}(t))' + \mathbf{G}(\mathbf{u}(t))\mathbf{G}(\mathbf{u}(t))',\tag{6}$$

$$\frac{\partial^2 \mathbf{f}}{\partial \mathbf{x}^2} \bullet \mathbf{P} = \left[\operatorname{tr}\left(\frac{\partial^2 \mathbf{f}_1}{\partial \mathbf{x}^2}\mathbf{P}\right), \operatorname{tr}\left(\frac{\partial^2 \mathbf{f}_2}{\partial \mathbf{x}^2}\mathbf{P}\right), \ldots, \operatorname{tr}\left(\frac{\partial^2 \mathbf{f}_N}{\partial \mathbf{x}^2}\mathbf{P}\right)\right]'$$

These ordinary differential equations are important to reformulate the initial problem as a deterministic optimal control problem involving only the mean and covariance of the original stochastic state process $\mathbf{x}_t$.

To do so, it must be noted that the expected cost function can also be rewritten in terms of the mean and covariance of $\mathbf{x}_t$ only as

$$C_{\text{ff}}(\mathbf{u}(t)) = r\,\Phi(\mathbf{m}(T), \mathbf{P}(T)) + \int_0^T l(\mathbf{m}(t), \mathbf{u}(t))\ dt\tag{7}$$

where $\Phi$ is a function of the terminal mean and covariance of the random variable $\mathbf{x}_T$. Note that the trajectory cost $l$ can be taken outside of the expectation because the control and mean are deterministic variables (by hypothesis and definition, respectively).

We thus obtain a deterministic optimal control problem (approximately equivalent to the stochastic problem defined by Eqs 3 and 4) to solve for the augmented state $(\mathbf{m}, \mathbf{P})$. This problem is summarized as:

$$\min_{\mathbf{u}(\cdot)} \left[r\,\Phi(\mathbf{m}(T), \mathbf{P}(T)) + \int_0^T l(\mathbf{m}(t), \mathbf{u}(t))\ dt\right],$$

$$\dot{\mathbf{m}} = \mathbf{f}(\mathbf{m}, \mathbf{u}) + \frac{1}{2}\frac{\partial^2 \mathbf{f}}{\partial \mathbf{x}^2}(\mathbf{m}, \mathbf{u}) \bullet \mathbf{P}, \quad \dot{\mathbf{P}} = \frac{\partial \mathbf{f}}{\partial \mathbf{x}}(\mathbf{m}, \mathbf{u})\,\mathbf{P} + \mathbf{P}\frac{\partial \mathbf{f}}{\partial \mathbf{x}}(\mathbf{m}, \mathbf{u})' + \mathbf{G}(\mathbf{u})\,\mathbf{G}(\mathbf{u})'\tag{8}$$

Interestingly, the efficient theoretical and numerical tools developed for DOC can be used to solve the above problem (e.g. [40]). Note that hard constraints for the final mean or covariance of the state can also be added in this formulation. We did so for the final mean state to ensure that the arm exactly reaches the desired target on average even though we could have modelled this constraint in the cost itself. The latter choice has the disadvantage of introducing additional tuning weights in the cost but could allow accounting for a terminal bias. Here we rather left the final covariance free because it was penalized in the cost function and was needed to determine an optimal movement duration without having to preset a desired amount of endpoint variance. The above optimal control problem can be run in free time, which means that the duration $T$ can be found automatically from the necessary optimality conditions of Pontryagin's maximum principle (instead of a laborious trial-and-error search process as it has been done in previous approaches such as [8] or [41]). The problem can also be run in fixed time, which means that the duration is preset by the researcher. We did so when investigating the evolution of optimal costs with respect to various movement durations

and to adjust noise magnitudes for fast or slow movements performed without vision (see Materials and methods section).

**Determining the feedback motor command via linear stochastic optimal feedback control.** During their execution, movements can be modified with incoming sensory information. This information can be exploited to form an optimal estimate of the current limb state, which can be used online through a linear locally-optimal feedback control scheme. Here, by linearizing the dynamics around the nominal expected state/control trajectories coming from SOOC, we will use the standard linear-quadratic-Gaussian framework [19, 37]. We shall also consider that, besides motor noise, there is some observation noise, the magnitude of which will depend on the available sensory modalities (e.g. with or without vision).

At this stage, we have access to a nominal open-loop control and an expected state trajectory, denoted by $\mathbf{u}(t)$ and $\mathbf{m}(t)$ respectively, for $t \in [0, T]$. We next extended the time horizon $T' > T$ in order to consider that executed movements have a longer duration than initially planned, assuming that the system is at rest for $t \geq T$.

To compute a locally-optimal feedback control, the dynamics is linearized around $\mathbf{m}(t)$ and $\mathbf{u}(t)$ using Taylor's expansions to obtain a linear-quadratic-Gaussian approximation in terms of state/control deviations (e.g. [42]) as follows:

$$d\mathbf{z}_t = \left( \mathbf{A}(t)\,\mathbf{z}_t + \mathbf{B}(t)\,\mathbf{v}_t \right) dt + \mathbf{G}(t)\,d\boldsymbol{\omega}_t \tag{9}$$

where

$$\mathbf{A}(t) = \frac{\partial \mathbf{f}}{\partial \mathbf{x}}\left(\mathbf{m}(t), \mathbf{u}(t)\right), \quad \mathbf{B}(t) = \frac{\partial \mathbf{f}}{\partial \mathbf{u}}(\mathbf{m}(t), \mathbf{u}(t)), \quad \mathbf{G}(t) = \mathbf{G}(\mathbf{u}(t)), \tag{10}$$

and

$$\mathbf{z}_t = \mathbf{x}_t - \mathbf{m}(t), \quad \mathbf{v}_t = \mathbf{u}_t - \mathbf{u}(t). \tag{11}$$

We further assume that the noisy sensory feedback $\mathbf{y}_t$ is obtained during motion execution from the following output equation:

$$d\mathbf{y}_t = \mathbf{g}(\mathbf{x}_t)\,dt + \mathbf{D}(t)\,d\boldsymbol{\eta}_t \tag{12}$$

where $\mathbf{y}_t \in \mathbb{R}^L$ and $\mathbf{g}(\mathbf{x}_t) \in \mathbb{R}^{L \times N}$ is the output function. The matrix $\mathbf{D}(t) \in \mathbb{R}^{L \times L}$ specifies how observation noise affects sensory feedback, where $\boldsymbol{\eta}_t$ is a $L$-dimensional standard Brownian motion process.

Using again a Taylor's expansion and defining $\mathbf{C}(t) = \frac{\partial \mathbf{g}}{\partial \mathbf{x}}(\mathbf{m}(t))$, the output equation can be approximated locally in terms of state deviations $\mathbf{z}_t$ as

$$d\mathbf{h}_t = \mathbf{C}(t)\,\mathbf{z}_t\,dt + \mathbf{D}(t)\,d\boldsymbol{\eta}_t \tag{13}$$

where $\mathbf{h}_t = \mathbf{y}_t - \mathbf{y}(t)$ with $\mathbf{y}(t) = \int \mathbf{g}(\mathbf{m}(t))dt$.

For this sub-problem, a quadratic cost function to ensure task achievement with minimal effort is defined as follows:

$$C_{\text{fb}}(\mathbf{v}_t) = \mathbb{E}\left[ \int_T^{T'} \mathbf{z}_t' \mathbf{R}\, \mathbf{z}_t\, dt + \int_0^{T'} \mathbf{v}_t' \mathbf{v}_t\, dt \right] \tag{14}$$

The locally-optimal feedback control law can be written as $\mathbf{v}_t = -\mathbf{K}(t)\,\hat{\mathbf{z}}_t$ where $\mathbf{K}(t)$ is the feedback gain matrix and $\hat{\mathbf{z}}_t$ is the optimal estimate of the state deviation $\mathbf{z}_t$ obtained from the

Kalman filter equation:

$$d\hat{\mathbf{z}}_t = (\mathbf{A}(t)\,\hat{\mathbf{z}}_t \,+\, \mathbf{B}(t)\,\mathbf{v}_t)\,dt + \mathbf{L}(t)(d\mathbf{h}_t - \mathbf{C}(t)\,\hat{\mathbf{z}}_t\,dt) \tag{15}$$

where $\mathbf{L}(t)$ is the optimal filter gain.

The problem defined by Eqs 9, 13 and 14 is a linear-quadratic-Gaussian problem, which can be solved using standard algorithms (e.g. [19]).

An overview of the SFFC model is given in Fig 2.

While the cost functions for the feedforward and feedback components both minimize error and effort terms (see Eqs (4) and (14)), they differ in several fundamental aspects. On the one hand, the feedforward cost function relies on deterministic variables that can be computed or estimated prior to the movement start. It aims at determining an optimal feedforward motor command, from which an expectation about the upcoming trajectory can be obtained. This cost minimizes effort (and possibly other terms such as smoothness) and endpoint variance, which in turn allows to specify the shape and characteristics of mean arm trajectories as well as the state covariance that would result from feedforward control (i.e. without online sensory feedback). Critically, this knowledge allows linearizing the arm's dynamics in order to apply the linear SOC framework subsequently. On the other hand, the feedback cost function depends on the stochastic deviations from the above expected control/state trajectories which will arise during movement execution. It ensures that the task will be achieved with a minimal amount of motor correction, in accordance with the minimal intervention principle [13]. This is done here by minimizing errors at the end of the movement (i.e. for times longer than the planned movement duration). The $\mathbf{R}$ term can be adjusted depending on the task, which will in turn determine the magnitude of the planned feedback gain $\mathbf{K}(t)$. To compute the feedback component of the motor command, related to online corrections, the system requires sensory information to update the estimate of the limb state throughout movement execution. Hence, internal or external perturbations inducing a deviation from the expected trajectory will be corrected via a task-dependent feedback mechanism. While SFFC assumes that the brain has some knowledge of the upcoming reach trajectory and feedforward command to reformulate the task in terms of state/control deviations, it must be noted that the feedback cost does not assume that a reference trajectory is tracked. If the task does involve trajectory tracking this can be handled by minimizing errors throughout the whole movement in the feedback cost or modelled by considering muscle viscoelasticity and mechanical impedance as in [33].

## Simulation results and comparison to experimental data

To test the SFFC model we consider a pointing task with a two-link arm moving in the horizontal plane from an initial posture to a target in Cartesian space (e.g. [31, 35, 36]). More details about the task, model of the arm and choice of parameters can be found in the Materials and methods section.

**Comparison to previous data of reaching movements without vision.** We first tested if SOOC can determine an optimal movement timing from the principle illustrated in Fig 1B. As SOOC does not consider the influence of online sensory feedback, its prediction would mainly correspond to the behavior of deafferented patients without vision of the moving hand (e.g. [43, 44]). It must be noted that SOOC at least requires an estimate of the initial arm's state to build the optimal feedforward motor command, in agreement with [43] who showed that prior vision of the arm improved movement precision in these patients.

Fig 3A illustrates that the evolution of the optimal expected cost with respect to movement duration. This U-shape cost function yields an optimal duration, which can be thought as the limit case of Fig 1B when observation noise is infinite. Remarkably, the resulting optimal

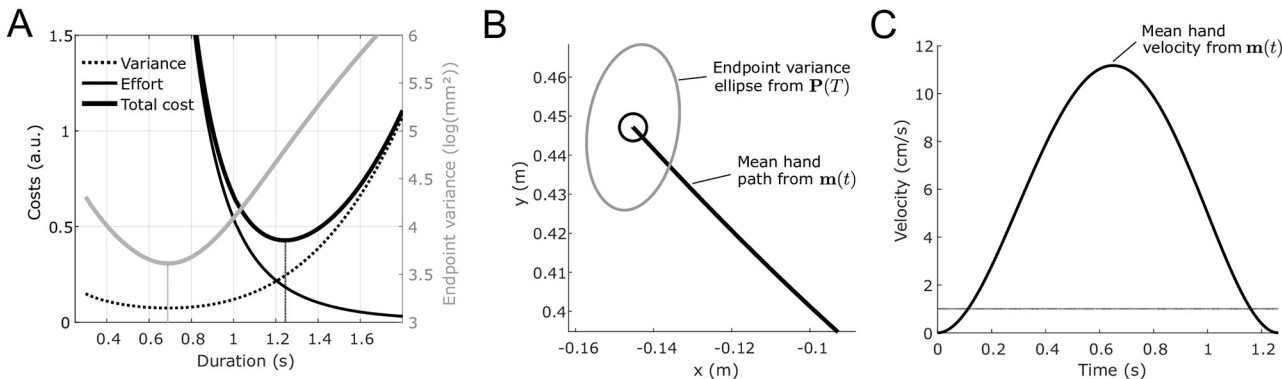

**Fig 3. Expected costs and trajectory predicted by SOOC for a horizontal point-to-point arm movement without feedback.** A. Evolution of the optimal costs with respect to the movement duration. The total cost function exhibits a U-shape, i.e. additive motor noise yields a minimal movement duration. The minimal duration (minimum of the black solid trace) is larger than the duration of minimum variance (the gray curve). B. Mean hand path and predicted endpoint variability (here depicted as a 90% confidence ellipse). These data can be computed from $\mathbf{m}(t)$ and $\mathbf{P}(t)$ respectively (converted from joint space to Cartesian space). The black circle depicts the target. C. Corresponding mean velocity profile. The dashed horizontal line indicates the threshold at which velocity profiles are cut in the experimental data. This figure is generated using simulation data of free-time optimal control computed with SOOC.

duration is longer than the duration of minimum variance, which is in agreement with the observation of [31]. Fig 3B and 3C show the corresponding mean hand trajectory, the path of which is approximately straight with bell-shaped velocity profile. This agrees with typical findings in healthy subjects and also with the overall strategy of deafferented patients without online vision [44]. Interestingly, SOOC also provides information about the final variability of the pointing under pure feedforward control (i.e. without feedback component; represented as a confidence ellipse in Fig 3B). The large final variability (compared to target size) is compatible with the relatively large endpoint variability exhibited by deafferented patients [44].

Next, we focus on previous experimental observations in healthy subjects performing movements without online visual feedback of the hand [31, 35, 36]. Healthy subjects typically have a smaller endpoint variability than deafferented patients without vision. In healthy subjects, proprioceptive feedback is indeed available and this sensory information can be used by the brain to build an estimate of the limb state. However, the absence of vision (and thus of multisensory integration) may degrade the hand state estimate [45], which can be accounted for in our model by assuming a relatively large observation noise in this case.

Fig 4A and 4B show the path of trials in the N-W and N-E directions when simulating the experiment of [31], where parameters were selected to reproduce the data in the N-W direction. The trajectories predicted by the model are similar to the trajectories experimentally measured in this task, with relatively straight hand path and bell-shaped velocity profile. The duration determined by the SFFC model was also in good agreement with the data. Interestingly, as in the experimental data of [31], the predicted duration was slightly longer for movements in the N-W than in the N-E direction, and the variance was also slightly larger in the N-W direction. It can also be noted that the endpoint variability is smaller with SFFC than with SOOC, thereby illustrating the improvement with proprioceptive feedback.

We then analyzed how the optimal movement duration depends on its direction and amplitude by using the same model parameters and still focusing on movements carried out without online visual feedback of the hand. Fig 4C–4F show how movements vary with the direction and with the distance. As in the experimental results of [35] and [36], movements in directions requiring more effort are slower. Fig 4F further shows that the predicted movement duration increases monotonically with the target distance as in experimental data [36].

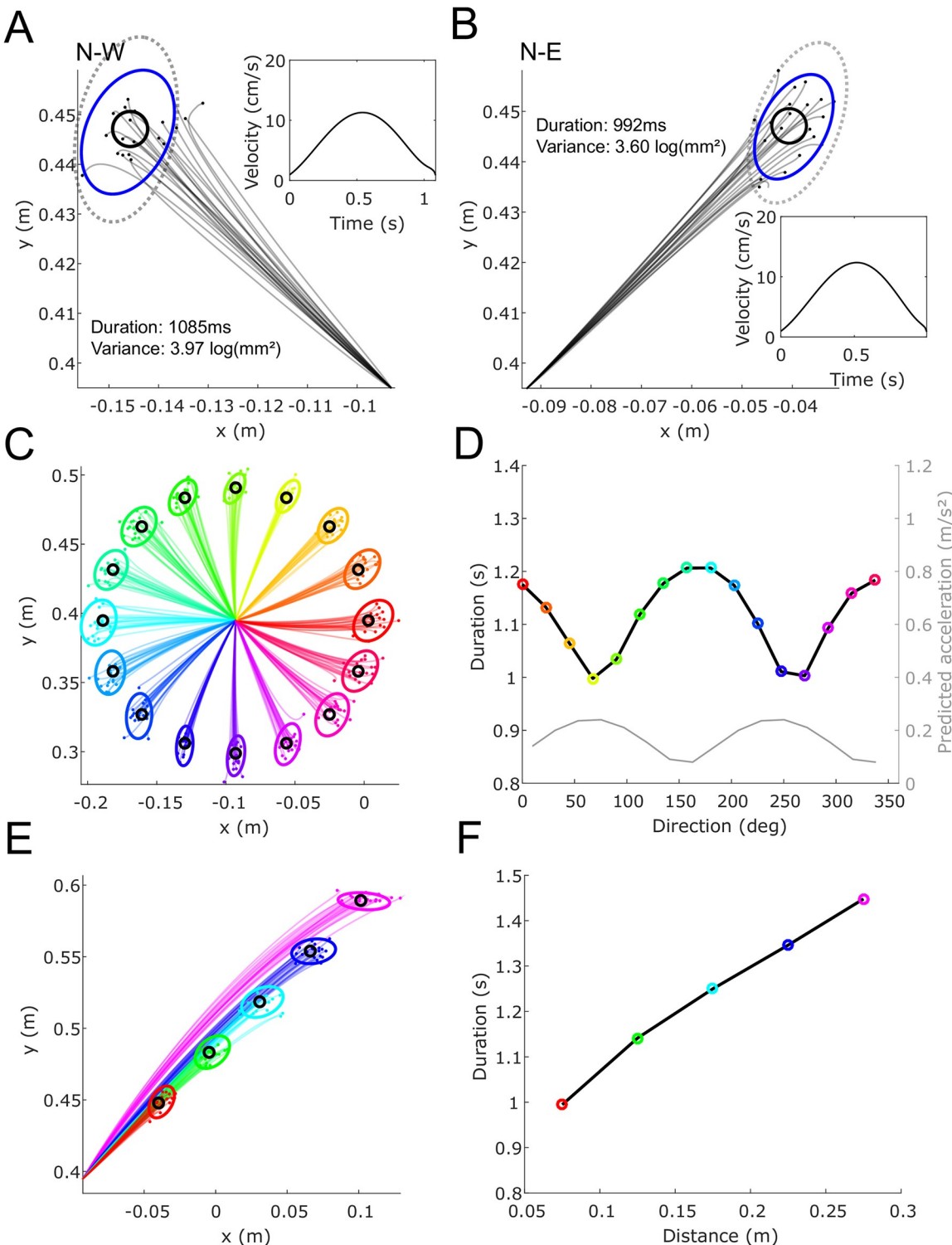

**Fig 4. Simulations of horizontal arm pointing movements without online vision of the hand.** A-B. Simulation of the data of [31]. Hand paths of 20 trials are shown in panels A and B for the N-W and N-E directions, respectively. 90% confidence ellipses of the end points are depicted in blue, which were computed from 1,000 samples. Dotted ellipses, corresponding to the endpoint variance of SOOC solutions, are depicted for comparison. The corresponding mean velocity profiles are also depicted as insets. The target is depicted as a black circle. Movement duration and endpoint variance are reported for each direction. C-F. Simulation of the data of [35, 36]. Hand paths of 20 trials are shown in panels C and E for the different directions and different distances for the N-E direction. 90% confidence ellipses of the end point are depicted (estimated from 1,000 samples). The corresponding durations are reported in panels D and F. In

panel D, the acceleration predicted from the hand mobility matrix is depicted (and calculated as in [36]). The direction-dependent and distance-dependent modulation of duration can be noticed.

These simulations suggest that the model can reproduce the basic characteristics of planar arm reaching movements without visual feedback, showing typical dependencies on distance and direction. Next, we compare movements carried out with and without vision. In particular, movements with vision are known to exhibit a smaller endpoint variability than analog movements without vision [45, 46].

**Comparison to data of reaching movements with and without vision.** We asked healthy participants to perform arm pointing movements to test the impact of online visual feedback of the hand on the preferred timing and variance of goal-directed movements. We wanted to estimate the extent to which movements with and without visual feedback differed in terms of timing and trial-by-trial variability, and whether these data could be replicated by the proposed SFFC model. Horizontal arm pointing movements of various directions {E, N-E, N-W, W} and amplitudes {0.06, 0.12, 0.18, 0.24} m both with and without vision of the moving hand (represented as a cursor on the screen, the actual arm being hidden) were recorded. Details about the task can be found in the Materials and methods section.

Fig 5A illustrates the experimental hand paths in all the directions and amplitudes in one participant. As expected, movements without vision were clearly less precise than movements with vision. This was confirmed by the group analysis reported in Fig 6A where the endpoint variance was computed for each distance and direction.

Two-way repeated measures ANOVAs confirmed a main effect of the visual condition ($F_{1,20}$ = 426.83, $p < 0.001$) with movements without vision exhibiting much more endpoint variance. A main effect of distance was also found ($F_{3,60}$ = 12.55, $p < 0.001$) and there was a

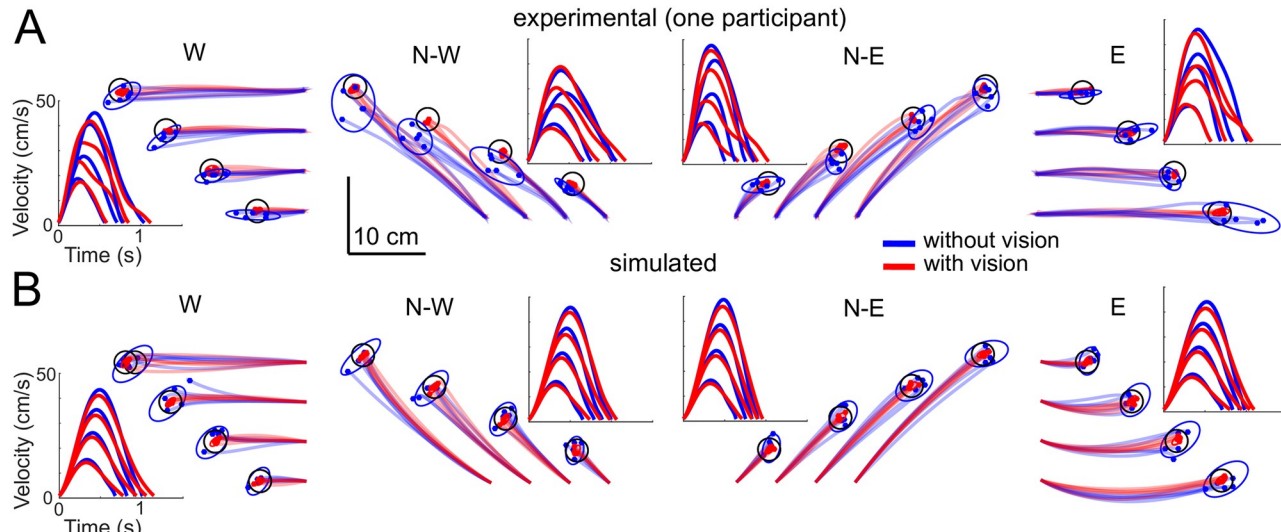

**Fig 5. Hand trajectories and velocities for an exemplar subject and for simulations.** Movements with vision and without vision are depicted in red and blue respectively. Panels A and B show experimental trajectories in the N-W and N-E directions respectively. The four distances {0.6, 0.12, 0.18, 0.24} m are represented by shifting the starting point for visibility. The real starting point was the same as described in Fig 8B. Targets are represented as 90% confidence ellipses for the endpoints, which were estimated from the five experiment's trials. Corresponding mean speed profiles over the five trials are computed after cutting the start and end using a 1.0 cm/s threshold and a time normalization. The same information is shown in panels C and D for 1,000 simulated movements with SFFC, where only 5 trajectories are depicted for clarity. The blue traces correspond to large observation noise and red traces to normal observation noise when vision is available. The simulation parameters were chosen to reproduce the average behavior of the participants, and not the plotted data of a specific participant.

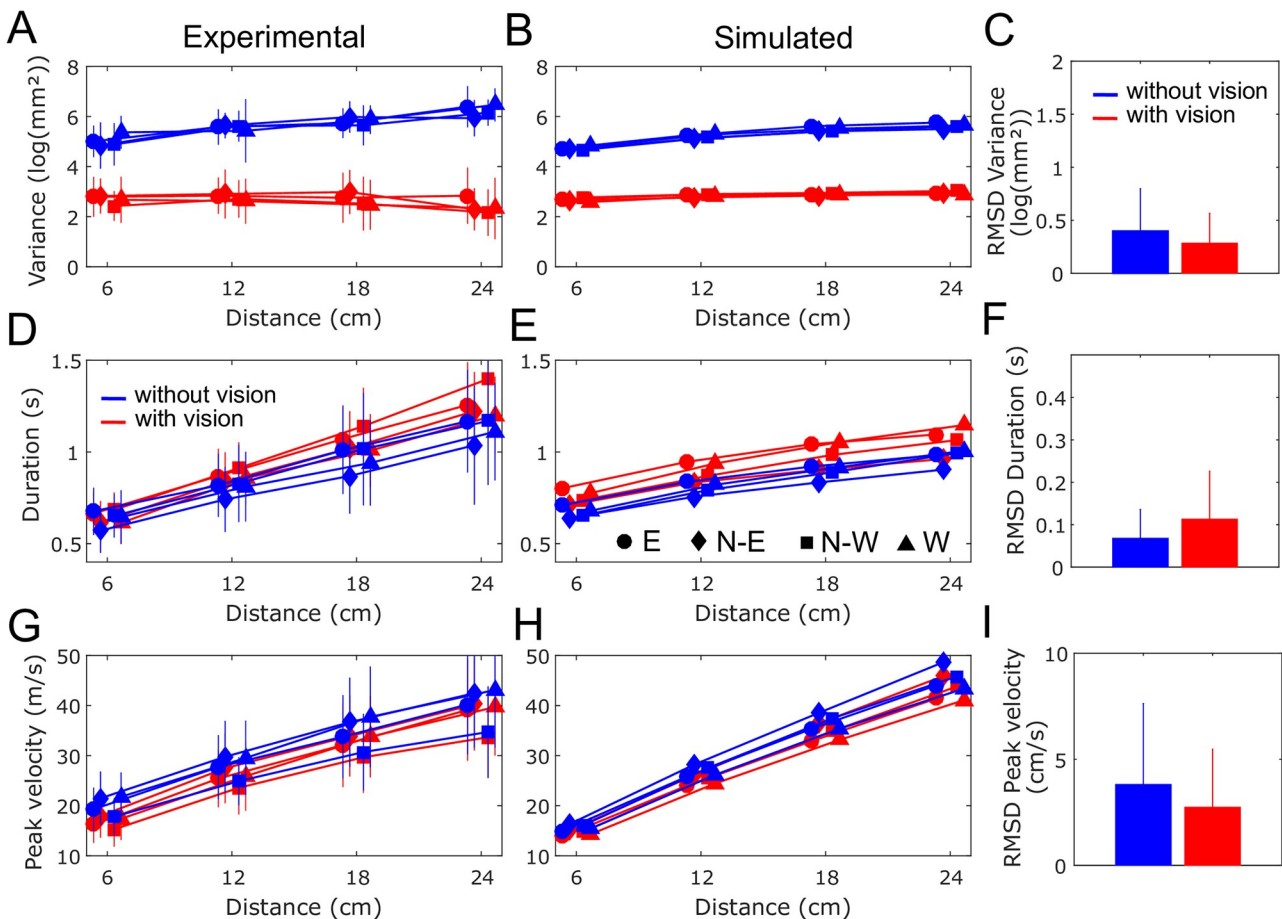

**Fig 6. Comparison of experimental and simulated data.** A. Mean experimental endpoint variance (across participants) for each direction and distance. Error bars indicate standard deviations across the 16 conditions (distance-direction pairs). Movement with and without vision are reported in red and blue, respectively. Circles, diamonds, squares and triangles represent the E, N-E, N-W and W directions, respectively. B. Same information for simulated movements. C. Root mean squared deviation (RMSD) between the real and simulated endpoint variance, (in log(mm²)). D-I. Same by reporting duration (in s) and peak velocity (in cm/s) instead of endpoint variance.

significant interaction ($F_{3,60} = 46.59$, $p < 0.001$) revealing that, without vision, participants were less and less precise as movement amplitude increased. A main effect of the direction was also detected on the variance ($F_{3,60} = 3.73$, $p < 0.05$), and there was no interaction effect between direction and condition ($p = 0.056$).

These empirical observations were well replicated by our model (Figs 5B and 6B). In particular, the increase of endpoint variance with distance is well predicted by the model with large observation noise and the gain of precision is also clear when vision is present and observation noise is thus reduced. To quantify the errors between the model predictions and the empirical data, we computed root mean squared deviations (RMSD). Fig 6C reports RMSD values averaged across all directions and distances for endpoint variance. The average RMSD was 0.40 and 0.28 log(mm²) for the without and with vision conditions respectively, which corresponded to 6.9% and 11.6% of the respective experimental mean values.

We next analyzed the timing of movements performed with and without vision (Fig 6D). A visual inspection reveals that the durations of movements with and without visual feedback of the hand exhibit similar trends, although movements with vision may tend to have slightly longer durations.

A two-way repeated measures ANOVA revealed no main effect of the visual condition on movement duration ($p = 0.055$). We found a main effect of distance ($F_{3,60} = 242.40$, $p < 0.001$) on movement duration (i.e. duration clearly increases with distance). A significant interaction effect between the visual condition and the distance was found ($F_{3,60} = 10.06$, $p < 0.001$). Post-hoc analyses revealed that only the 24 cm distance had significantly longer duration with vision compared to without vision ($p = 0.014$). Regarding the effect of direction on duration, a significant interaction effect between the visual condition and direction was found ($F_{3,60} = 5.66$, $p = 0.002$). Post-hoc analyses mainly revealed that N-W movements with vision lasted longer than the other directions of movement ($p < 0.001$). The model replicated the increase of movement duration with distance relatively well, although some variations with respect to direction were less clear in these data (see Fig 6E). Quantitative comparisons are reported in Fig 6F and reveal that, on average across distance and direction conditions, RMSD for duration was 70 and 113 ms for the without and with vision conditions respectively, which corresponded to 7.1 and 11.8% of the respective experimental mean values.

To analyze the differences in movement timing with a variable less sensitive to terminal adjustments, we repeated the above analyses using peak velocity instead of duration (Fig 6G). We found neither a main effect of the visual condition ($p = 0.052$) nor an interaction effect ($p = 0.262$) with distance. Although there was a trend to have slightly lower peak velocities with vision, no statistical difference was observed on peak velocity for movements with and without vision (even for the largest distance, 24 cm, in contrast to the results found for duration). A main effect of distance on peak velocity was found as expected since peak velocity clearly increases with movement distance ($F_{3,60} = 253.72$, $p < 0.001$). Regarding the effect of direction, we found a significant interaction ($F_{3,60} = 2.83$, $p < 0.05$). Post-hoc tests mainly indicated that N-W movements were slower than those in other directions with and without vision ($p < 0.01$). The model replicated well the increase of peak velocity with distance (Fig 6H) and the dependence of peak velocity on direction was again less clear in these data. RMSD for peak velocity was on average 3.8 and 2.7 cm/s for the without and with vision conditions respectively, which corresponded to 13.4 and 10.0% of the respective experimental values (Fig 6I).

Finally, a correlation analysis was carried out to analyse the extent to which the timing properties of movements performed with and without vision were related (Fig 7A for durations and Fig 7B for peak velocities). We found strong correlations in experimental data ($R^2 > 0.96$), thereby confirming the consistency of movement timing with and without visual feedback. Similarly strong correlations were found in simulated data based on the proposed model. The main reason is that, in the model, movements with or without vision are both based on the same feedforward motor command and just differ here in the magnitude of observation noise (which was assumed to be ×10 larger for movements without vision than for movements with vision).

## Discussion

This paper introduced the stochastic optimal feedforward-feedback control model (SFFC) of learned goal-directed arm movements unifying previous optimal control models that focused either on deterministic or stochastic aspects of movement. The SFFC suggests how the nervous system may cope with noise and delays by combining feedforward and feedback motor command components. It can be used to predict the nominal timing and variability of reaching movements with degraded sensory feedback, as was illustrated on movements carried out without visual feedback. We discuss below the main aspects of this new model in perspective with experimental results and previous models from the literature.

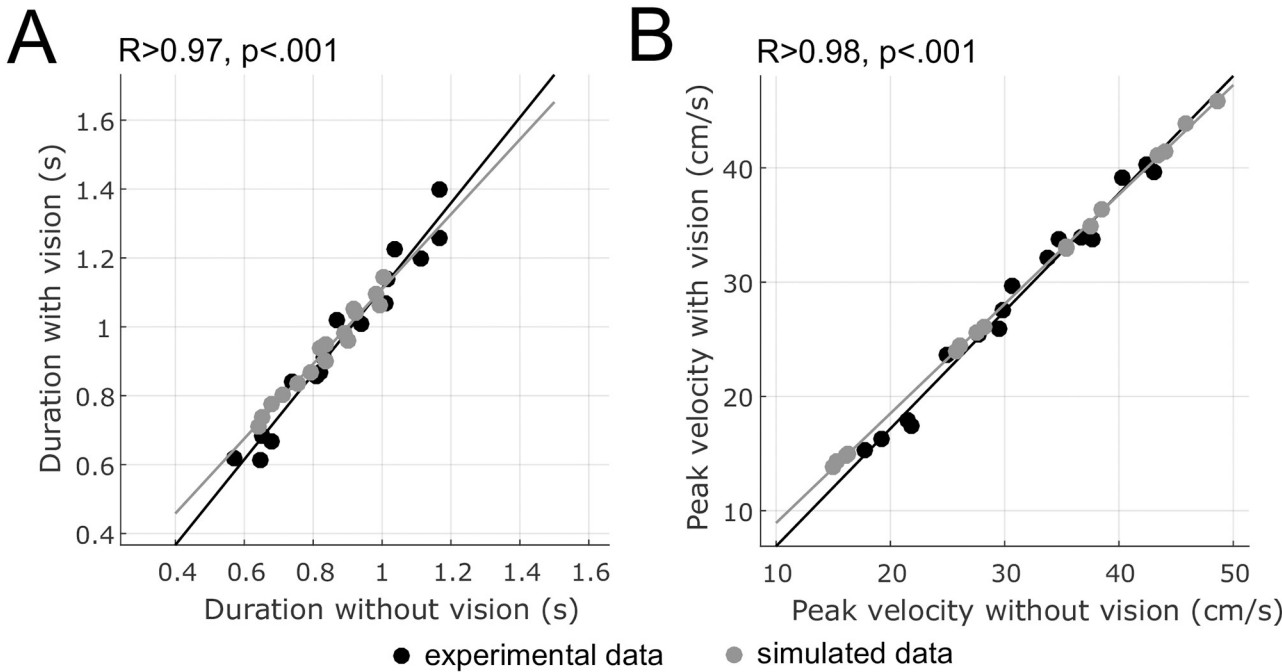

**Fig 7. Correlations of duration or peak velocity between movements with and without vision.** A. Correlations for durations. Each data point represents one condition of distance and direction (averaged across participants). Experimental and simulated data are plotted respectively in black and grey. B. Correlations for peak velocities. Regression lines are plotted for the illustration.

### How existing models predict movement timing and variability

The development of SFFC was prompted by the difficulty to predict movement timing independently of endpoint variability with existing optimal control models. Optimal control being a versatile framework to model human motor control [47, 48], several classes of models have been proposed with prediction of movement timing and variability summarized in Table 1. Seminal deterministic optimal control (DOC) models can predict the shape of average arm trajectories corresponding to a given movement duration [17, 18, 49, 50]. The movement duration can be determined in ad hoc ways such as by setting the task's effort [51–53] but DOC does not account for the trial-by-trial variability of human movement. Assuming signal-dependent motor noise, SOOC models have been proposed to extend deterministic models and predict a movement duration corresponding to a fixed level of endpoint variance (e.g. the width of the target) [8, 20], but these models will follow Fitts' law [54], which does not hold for

**Table 1. Predictions of movement timing (duration or speed) and endpoint variance (variability across trials) with different types of optimal control models.** Some ad hoc fixes have been introduced in some models to predict timing and/or variability, unlike the SFFC model.

| Type | Timing | | Variability | | Models |
|---|---|---|---|---|---|
| | vision | no vision | vision | no vision | |
| DOC | N | N | N | N | [17, 18, 49] |
| DOC with cost of time or other fixes | Y | Y | N | N | [24, 25, 51–53, 56] |
| SOOC | Y | Y | N | Y | [8, 20, 34] |
| SOC | N | Y | Y | Y | [14, 19] |
| SOC with cost of time or other fixes | Y | Y | Y | Y | [22, 41] |
| SFFC | Y | Y | Y | Y | introduced in this paper |

self-paced arm movements [55]. Here we showed that SOOC can explain the timing and variability of self-paced movements carried out without sensory feedback by considering the effects of motor noise together with a minimum effort-variance cost. However, SOOC will not account for the drastic reduction of variability exhibited by movements executed with proprioceptive and/or visual feedback. Muscle co-contraction and mechanical impedance that could be modelled as in [34] may contribute to reduce this variability but not to the level of movements with online multi sensory feedback.

SOC emphasized the role of high-level feedback to reliably execute a motor task despite relatively large variability in repeated movements. In SOC, the motor command is a function of a limb state estimate built from internal dynamic predictions and delayed sensory information. By considering sensorimotor noise, and minimizing error and effort [13, 14, 19], these models correct task-relevant errors according to the minimal intervention principle. However, as the expected cost typically plateaus for visually-guided movements of long duration (see Fig 1A), SOC cannot predict a finite movement duration without ad hoc criterion. For instance, [22] determined duration in an infinite-horizon SOC formulation by comparing the magnitude of endpoint variance to the target's width, which allowed to predict the speed-accuracy trade-off. To model the variability of movements without vision, SOC models will normally assume a large observation noise. This will degrade limb state estimates and make the controller more dependent on internal predictions corresponding to a feedforward mechanism. The fact that movements carried out with and without vision had a highly correlated timing in our experimental data is supporting the hypothesis that these two types of movement have a common origin, which can be captured by a feedforward motor command. The same conclusion was drawn by [45] who found a reduction of feedback gains in a reaching task when visual feedback was removed. The authors suggested that a feedforward motor command is needed to explain that movements with lower feedback gains had well preserved kinematics and timing. Note however that visually-guided movements have a tendency to be slower likely due to the integration of visual corrections at the end of the movement, to adjust more accurately the final cursor's location. This is consistent with the observation that in our experiment the peak velocities with and without vision were even more similar than movement durations. Overall, SFFC appears as the first model that can explain timing and variability of arm movement trajectories carried out with or without visual feedback from neuromechanic considerations only.

## Is movement timing due to neuromechanic or neuroeconomic factors?

Previous arm movement models [24, 25, 56] used a cost of time to limit the movement duration. In these DOC models, the cost-of-time parameters to accurately reproduce the movement timing observed experimentally can be determined using inverse optimal control techniques [25, 27]. By extension, SOC with a cost of time has been used to model saccades [41] and, in principle, SOC with a cost of time may be able to reproduce the above experimental results. However, it is not straightforward to find an optimal duration in SOC from computing the cost for all possible durations, and to adapt such models to the nonlinear dynamics of the human arm. In contrast, it would be straightforward to include a cost of time in SFFC (in the term $l(\mathbf{x},\mathbf{u})$) and to use it to determine the optimal movement duration from necessary optimality conditions. However our objective here was to understand if neuromechanical factors could explain the timing and variability of self-paced arm movements, which led us to develop the SFFC model. While above simulations and experimental results showed that SFFC could explain the timing of simple pointing movements toward targets without ad hoc hypothesis, further investigations would be required to determine whether it could also account for individual differences [27, 57–59] and sensitivity to reward [60–62], e.g. by varying the factor $r$

to trade-off variance and effort for instance. Experiments may also be developed to test the model's prediction that movement timing should increase with larger constant motor noise and decrease with larger signal-dependent motor noise.

### The role of motion planning and feedforward control

Overall, this study suggests the importance of motion planning in the generation of goal-directed arm movements. A large body of experimental evidence has shown the critical role of motion planning to select a suitable motor solution for carrying out a task (see [4] for a review). The picture suggested by previous studies and above modeling is that the CNS executes well-learned, unperturbed movements using an important feedforward component to the motor command, given the intrinsic noise, delays and task dynamics. The sensorimotor plans required for such control strategy may be learned by gradually minimizing reflexes and integrating voluntary (e.g. visual) corrections after movement [63–65]. This learning will minimize the reliance on high-level feedback corrections to achieve the task and thus gradually incorporate in the feedforward motor command any feature that can be identified over trials.

The behaviour after learning could be captured by the SFFC model that integrates feedforward and feedback control. The simulation results illustrated how SFFC combines the advantages of SOOC [8, 33, 34] and SOC [10–12] to explain the timing and the variability of arm movements performed with or without visual feedback of the moving limb, by minimizing the consequences of signal-dependent and constant motor noise on endpoint variance as well as effort or kinematic costs such as smoothness. One important aspect of SFFC is that the feedforward motor command already considers uncertainty about the task dynamics (e.g. motor noise or unknown perturbations, like in [8]) and can incorporate this knowledge in the plan to adjust the mechanical impedance to the task's uncertainty [34, 66, 67]. This feedforward motor command is complemented by a high-level feedback motor command that corrects task-relevant deviations resulting from perturbations not handled by the feedforward motor command, such as accumulation of positional errors due to constant noise [31, 68], visually elicited corrections [64, 69] or long-latency proprioceptive feedback responses to large mechanical perturbations [70, 71]. The state-feedback gain is also part of the motor plan, the magnitude of which can be adapted depending on the task (by tuning the weights in the feedback cost function in SFFC). This general planning scheme highlights how feedforward motor commands (which determine the nominal shape, timing and variability of unperturbed trajectories) and feedback motor commands (which handle the corrections of task-related errors using current limb state estimates) could yield a skillful motor control strategy.

## Materials and methods

### Ethics statement

The experimental protocol was approved by the Université Paris-Saclay local Ethics Committee (CER-Paris-Saclay-2019–031). Written informed consent was obtained from each participant prior to starting with the experiment.

### Experimental task and procedures

**Participants and experimental setup.**   21 young adults (24.5 ± 2.0 years old [mean±std], height 1.74 ± 0.09 m, with 11 females and 4 left-handed) participated in this study. All participants had normal or corrected to normal vision, and no known neurological impairment or mental health issue. Each participant was seated, and had to move a stylus on a Wacom tablet

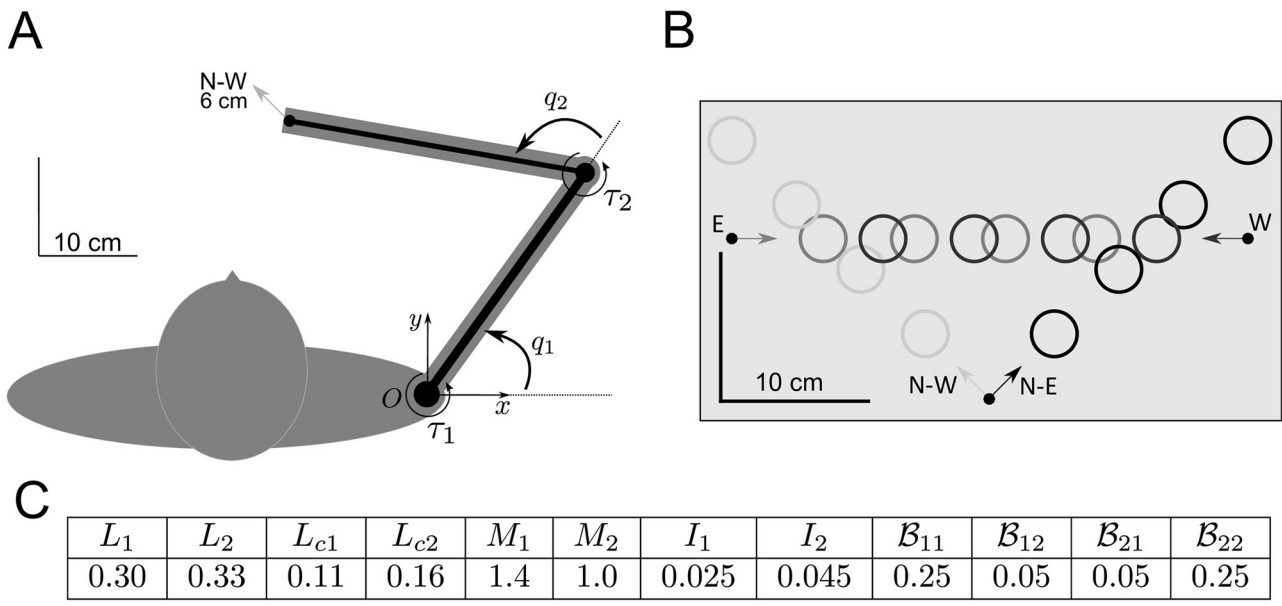

**Fig 8. Arm reaching task in our experiments and simulations.** A: Two-link model of the arm and planar movement used to set the model parameters. B: Horizontal arm movements carried out with and without online vision of the hand (4 directions, W, N-E, N-W and E, and 4 distances, 6, 12, 18 and 24 cm) in our experiment. C: Arm parameters used in the simulations.

(Wacom Intuos 4 XL) laid on horizontal table. The location of the stylus on the tablet was displayed on a monitor placed in front of the participant (i.e. on a vertical screen).

**Pointing task in two conditions: With and without online visual feedback.**   When a participant was ready, a 5 mm diameter disk appeared on the screen indicating the start position, on which they was instructed to move the cursor. Once the center of the cursor was within the start disk for 1 second, this was replaced by a 3 cm diameter target disk placed at 6, 12, 18 or 24 cm from the start position. Reaching movements were carried out in four directions as indicated in Fig 8A and 8B. If the start position was at the bottom of the screen ($x$-$y$ coordinates with respect to the shoulder [-15, 30] cm), the target was placed in the N-E or N-W direction. If the start was on the left of the screen (coordinates [-29, 34.5] cm, it was in the E direction, and if it was on the right of the screen (coordinates [5, 34.5] cm) in the W direction. This resulted in 16 different possible movement types.

The participants were instructed to move the cursor at comfortable pace in order to reach the target, without leaning the arm on the tablet. Note that their arm was hidden by a cardboard box so that they could not see it. They had to perform reaching movements either without or with the cursor displaying their hand position on the screen during the movement. In the non-visual condition, the cursor disappeared at the beginning of the movement and reappeared 1 s after the end of the movement, to indicate the pointing error and thus avoid the endpoint to gradually drift trial after trial.

Each participant started with a familiarization phase of 32 pointing movements including 16 consecutive trials per condition (with and then without visual feedback). Then they had to perform 160 trials, with 80 per visual feedback modality. The different movement types and the visual modalities were presented in pseudo-random order. This resulted in 5 trials of each amplitude and direction for each starting position. Every 10 trials a break of approximately 1 minute was scheduled, during which they could place the forearm on the tablet or the desk.

**Data acquisition and parameters of interest.**   The stylus position was recorded at 125 Hz with MATLAB (The MathWorks, Inc.), and the Psychtoolbox [76] was used to display the

stimuli on the screen. The system was calibrated so that a movement of the stylus on the tablet corresponded to a movement of the same length of the cursor on screen. The raw data were smoothened for further analysis using a 5th-order Butterworth low-pass filter with 12.5 Hz cutoff frequency and without delay. Velocity was computed via numerical differentiation. Among the parameters of interest, we computed the movement duration using a velocity threshold of 1 cm/s, the peak velocity of the maximal value of velocity profiles (in cm/s), and the endpoint variance (in $\log(\text{mm}^2)$). In every trial, the movement's endpoint was determined by the last recorded position at the end of the movement time. Endpoint variance was then estimated from the trace of the covariance matrix of final positions and the logarithm of this value was computed as in [31].

**Statistical analysis.** Two-way repeated measures ANOVAs with condition (with vision and without vision) and amplitude (from 6 to 24 cm) or direction (from E to W) as within-subjects factors were carried out to assess the variation of movement timing (i.e. duration and peak velocity) and variance across conditions. Moreover, a correlation analysis was performed to assess the relationships between the timing of movements carried out with and without vision.

## Numerical simulations

Arm reaching movements were simulated using a 2-link arm model with joint configuration vector

$$\mathbf{q} = \begin{bmatrix} q_1 \\ q_2 \end{bmatrix} \tag{16}$$

where $q_1$ is the shoulder and $q_2$ the elbow angles. The skeletal dynamics of the arm was described by the rigid body model of Eq (1) with:

$$\mathcal{M}_{11}(\mathbf{q}) = I_1 + I_2 + M_2 L_1^2 + 2 M_2 L_1 L_{g2} \cos(q_2), \quad \mathcal{M}_{12}(\mathbf{q}) = I_2 + M_2 L_1 L_{g2} \cos(q_2),$$

$$\mathcal{M}_{21}(\mathbf{q}) = \mathcal{M}_{12}(\mathbf{q}), \quad \mathcal{M}_{22}(\mathbf{q}) = I_2, \quad \mathcal{C}_{11}(\mathbf{q}, \dot{\mathbf{q}}) = -2 M_2 L_1 L_{g2} \sin(q_2) \dot{q}_2,$$

$$\mathcal{C}_{12}(\mathbf{q}, \dot{\mathbf{q}}) = -M_2 L_1 L_{g2} \sin(q_2) \dot{q}_2, \quad \mathcal{C}_{21}(\mathbf{q}, \dot{\mathbf{q}}) = M_2 L_1 L_{g2} \sin(q_2) \dot{q}_1, \quad \mathcal{C}_{22}(\mathbf{q}, \dot{\mathbf{q}}) = 0.$$

For the planar movements considered in this paper the gravity term $\mathcal{G}$ is set as zero. $\{I_i, L_i, L_{gi}\}$ and $\{M_i\}$ are the moments of inertia, lengths of segments, lengths to the centre of mass and mass of the segments.

Furthermore, we have

$$\mathbf{f}(\mathbf{x}_t, \mathbf{u}(t)) = \begin{bmatrix} \dot{\mathbf{q}}_t \\ \mathcal{M}^{-1}(\mathbf{q}_t)[\boldsymbol{\tau}_t - \mathcal{C}(\mathbf{q}_t, \dot{\mathbf{q}}_t)\dot{\mathbf{q}}_t - \mathcal{B}\dot{\mathbf{q}}_t] \\ \mathbf{u}(t) \end{bmatrix}, \quad \mathbf{G}(\mathbf{u}(t)) = \begin{bmatrix} 0 & 0 & 0 & 0 \\ 0 & 0 & 0 & 0 \\ 0 & 0 & 0 & 0 \\ 0 & 0 & 0 & 0 \\ \sigma_1 & d_1 u_1(t) & 0 & 0 \\ 0 & 0 & \sigma_2 & d_2 u_2(t) \end{bmatrix}, \tag{17}$$

where the parameters $\{\sigma_i\}$ are used to set the magnitude of additive noise and $\{d_i\}$ the magnitude of multiplicative noise.

Regarding the feedforward cost function, we define $\phi(\mathbf{m}(T), \mathbf{x}_T)$ to estimate the covariance of the final hand position. Denoting by $\mathbf{J}(\mathbf{q})$ the Jacobian matrix of the two-link arm, an approximation of this function can be computed by:

$$\phi(\mathbf{m}(T), \mathbf{x}_T) = \text{tr}([\mathbf{J}(\mathbf{m}_q(T))(\mathbf{q}_T - \mathbf{m}_q(T))][\mathbf{J}(\mathbf{m}_q(T))(\mathbf{q}_T - \mathbf{m}_q(T))]') \tag{18}$$

where $\mathbf{m}_q(T)$ is the mean final position of the random variable $\mathbf{q}_T$ (i.e. the 2-dimensional vector of final joint positions) and tr denotes the trace of the matrix. The expectation of $\phi(\mathbf{m}(T), \mathbf{x}_T)$ can then be rewritten as a function of the mean and covariance of the state process $\mathbf{x}_t$:

$$\Phi(\mathbf{m}(T), \mathbf{P}(T)) = \text{tr}[\mathbf{J}(\mathbf{m}_q(T))\mathbf{P}_q(T)\mathbf{J}'(\mathbf{m}_q(T))] \tag{19}$$

where $\mathbf{P}_q$ is the 2×2 covariance matrix of joint positions.

The infinitesimal cost $l(\mathbf{m}, \mathbf{u})$ is defined as follows:

$$l(\mathbf{m}, \mathbf{u}) = \mathbf{u}'\mathbf{u} + \alpha(\ddot{x}^2 + \ddot{y}^2) \tag{20}$$

where $x$ and $y$ denote the mean Cartesian positions of the hand (which can be computed from $\mathbf{m}(t)$ and the forward kinematic function). This cost implements a compromise between effort (here measured as squared torque change) and smoothness (here squared hand jerk) through the $\alpha$ parameter. Evidence for composite cost function mixing kinematic and dynamic or energetic criteria has been found in previous works [50, 72]. The jerk term is useful to correct for abnormal asymmetries in velocity profiles which may arise partly from the minimum torque change model (e.g. [73]), but this term does not affect our results otherwise.

For the linear-quadratic-Gaussian sub-problem, we set $\mathbf{C} = \mathbf{I}_6$ (identity matrix) meaning that we assume that both position, velocity and force could be estimated from multisensory information as in [19]. We verified that the same results and conclusions were obtained by limiting the observation matrix to the position and velocity components only. The observation noise matrix $\mathbf{D}$ was taken of the form $\mathbf{D} = \beta \mathbf{I}_6$ where $\beta$ specifies the overall magnitude of observation noise. This parameter can be varied depending on whether vision of the cursor is available or not during the movement. Finally, for the feedback cost function, we set $\mathbf{R} = \rho\text{diag}(1, 1, 0, 0, 0, 0)$ such that only the deviations about the final arm posture defined by the target location were penalized during the post-movement interval.

The SOOC solutions were obtained with the $\mathcal{GPOPS}$ optimal control software that approximates the continuous-time optimal control problem as a sparse nonlinear programming problem [40]. To compute the SFFC solutions, we considered a discrete time approximation of the linear-quadratic-Gaussian sub-problem around the SOOC solution with a time step of $dt = 0.005$ s. Standard discrete-time algorithms for linear-quadratic-Gaussian control were then used to compute the gains [19]. In our simulations, we extended the time horizon by 1 s ($T' = T+1$) to consider movements longer than the planned duration $T$. We tested different extended horizon between 0.5 s and 2 s and it did not change the results. It is worth noting that sensory feedback delays can be easily handled at this stage due to the discrete time approximation. All the simulations were performed with MATLAB (Mathworks, Natick, MA).

**Selection of model parameters.**   The arm parameters used in the simulations (from [42], in SI units) are given in Fig 8C.

The remaining parameters of the model are related to cost functions ($\alpha$, $r$ and $\rho$) and noise magnitudes ($\{\sigma_i\}$, $\{d_i\}$ and $\beta$). Some of these parameters affect the design of the feedforward command ($\alpha$, $r$, $\{\sigma_i\}$, $\{d_i\}$) and the others affect the design of the feedback command ($\rho$, $\beta$). First, we verified that the qualitative predictions and principles of the model were robust to parameters choices. Second, to have simulations that correspond quantitatively to experimental data, we adjusted the parameters using the procedure described hereafter. Note that we did

not try to find the best-fitting parameters using an automated procedure but adjusted the parameters to yield timing and variance of the same order of magnitude as experimental data.

We first fixed $\alpha = 0.02$ in all simulations, to implement a compromise between torque change and hand jerk. Note that we also considered $\alpha = 0$ and the results revealed that the smoothness term contributes to get slightly more linear hand paths with more bell-shaped velocity profiles, but this does not affect the main findings. Second, to reduce the number of parameters, we assumed that the magnitude of additive and multiplicative motor noise are the same in the two joints of the arm, i.e. $\sigma_1 = \sigma_2 = \sigma$ [rad/s$^{3/2}$] and $d_1 = d_2 = d$ [rad/(Nm s$^{1/2}$)]. The three remaining free parameters for SOOC ({$\sigma, d, r$}) were then adjusted by considering a movement of 7.4 cm in the N-W direction by using the existing data of [31] as a reference. The initial arm configuration was approximately $q_1(0) = 50°$ and $q_2(0) = 100°$ in this experiment. In the N-W movement, both joint angles change significantly, so that the effects of noise magnitude can be estimated in the two degrees of freedom using the three steps as follows:

- Since additive noise dominates at low speed, the magnitude of constant noise was adjusted on 1400 ms long movement in order to obtain an endpoint variance larger than what has been found in [31]. Indeed, these data were obtained for movements without vision in healthy subjects, where proprioceptive feedback was still available, and analog movements in deafferented patients would exhibit a larger endpoint variance [43, 44]. This resulted in $\sigma = 0.005$ [rad/s$^{3/2}$] and in about 6 log(mm$^2$) of endpoint variance (which is larger than the 4.2 log(mm$^2$) measured in [31]).

- Since multiplicative noise dominates for fast speed movements, which are less affected by feedback, multiplicative noise was adjusted on 350 ms long movements based on the data of [31]. $d = 0.01$ yielded an endpoint variance about 4.1 log(mm$^2$).

- The variance weight $r$ was then adjusted to fit the preferred duration of movements observed in the N-W direction. We found that $r = 2,000$ yields a movement time of about 1080 ms, which is similar to the preferred duration in [31].

Once the SOOC solution was obtained, we determined the remaining parameters of the SFFC model, which are related to the linear-quadratic-Gaussian sub-problem. We first set the observation noise $\beta$ and feedback cost weight $\rho$ by assuming that visually-guided movements are performed accurately at the preferred speed. This resulted in $\beta = 0.003$ and $\rho = 1,000$ for the data of [31]. Without vision, only proprioceptive feedback can be used and we assume that this leads to an increase of sensory variance. This increase was chosen to match the endpoint variance observed without vision in [31] (<4 log(mm$^2$)), and this led to $\beta = 0.03$ (i.e. ×10 larger than the magnitude of observation noise with vision). Note that we did not change $\rho$ in the present simulations, but we also considered that the product $\rho\beta$ could remain constant (i.e. $\rho = 100$ if $\beta = 0.03$), which reduced the feedback gain without affecting much the simulations for the unperturbed reaching movements under consideration.

When simulating movements without vision (or without feedback at all), a basic stopping mechanism was added by increasing joint friction 50 ms before the planned movement end to ensure that the terminal velocity always falls below the threshold (we added 3.5 kg m$^2$/s to $\mathcal{B}_{ii}$, i = 1, 2), which corresponds to a larger muscle viscosity at low speed [74]. Note that to compare simulated and experimental durations, we systematically applied a 1 cm/s threshold on hand velocity in agreement with experimental data processing (see above and [31]).

This set of parameters was used to simulate movements of different durations and directions, and compare the predictions to existing data. We used Eq (9) and the above parameters to generate reaching movements of duration 300–1450 ms in the N-W and N-E directions, and then computed the different optimal costs. Next, we tested the model predictions by

computing optimal movement durations for increasing distances ({7.5, 12.5, 17.5, 22.5, 27.5} cm in the N-E direction), and in eight directions as in classical experiments of arm reaching movements without vision [35, 36].

**Simulation of new experiment with SFFC.**   To simulate movements with and without vision described in Fig 8A and 8B, two previous parameters had to be adjusted to account for the larger variability and the shorter durations observed in our data compared to the experiment of [31]. This adjustment was made to have a better quantitative fit of the experimental data but the qualitative results would be the same if keeping previous parameters unchanged. These changes may be due to differences in experimental protocols (target's width, arm's weight support, instructions etc.). Therefore, to reflect larger variance and shorter durations, we set $\sigma = 0.025$ and $r = 6,000$ and kept the other parameters invariant. Here, to also investigate the influence of sensory delays, we performed simulations by considering a 50-ms delay in sensory feedback loops. This was done in the discrete-time approximation of the linear-quadratic-Gaussian sub-problem by using the classical procedure consisting of augmenting the system's state to include delayed instances of the state process (e.g. see [75] for details). Note that delays did not affect much the present simulations. This was verified by simulating SFFC with and without delays and very similar quantitative results were obtained for the tested movements. We report the simulations for the delayed case.

## Supporting information

**S1 Data. Experimental data supporting Fig 6.**
(XLS)

## Author Contributions

**Conceptualization:** Bastien Berret, Nicolas Schweighofer, Etienne Burdet.

**Formal analysis:** Bastien Berret, Nicolas Schweighofer, Etienne Burdet.

**Investigation:** Bastien Berret, Adrien Conessa, Nicolas Schweighofer, Etienne Burdet.

**Methodology:** Bastien Berret, Adrien Conessa, Etienne Burdet.

**Software:** Bastien Berret, Adrien Conessa.

**Supervision:** Bastien Berret.

**Validation:** Bastien Berret, Etienne Burdet.

**Visualization:** Bastien Berret, Adrien Conessa.

**Writing – original draft:** Bastien Berret, Adrien Conessa, Etienne Burdet.

**Writing – review & editing:** Bastien Berret, Adrien Conessa, Nicolas Schweighofer, Etienne Burdet.

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
