## [Decision Letter · Decision Letter 0]

24 Jan 2021

Dear Dr. Berret,

Thank you very much for submitting your manuscript "Stochastic optimal feedforward-feedback control determines timing and variability of arm movements with or without vision" for consideration at PLOS Computational Biology.

As with all papers reviewed by the journal, your manuscript was reviewed by members of the editorial board and by several independent reviewers. In light of the reviews (below this email), we would like to invite the resubmission of a significantly-revised version that takes into account the reviewers' comments.

While we have invited a resubmission, as you will see below some of the reviewers were conflicted about how to treat this paper. While there was general praise for the methodological aspects of the modeling, reviewers 1 and 2 especially questioned the assumptions that went into the modeling, such as considering reaching without vision as open loop control.

It is not clear whether this criticism can be addressed, but if you believe that is possible we would be happy to consider the paper again.

----

We cannot make any decision about publication until we have seen the revised manuscript and your response to the reviewers' comments. Your revised manuscript is also likely to be sent to reviewers for further evaluation.

Sincerely,

Ulrik R. Beierholm

Associate Editor

PLOS Computational Biology

Samuel Gershman

Deputy Editor

PLOS Computational Biology

Reviewer's Responses to Questions

**Comments to the Authors:**

Reviewer #1: The authors present a model of human upper limb control composed of a feed-forward specification of a control sequence determined by the minimization of variance and effort, coupled with a local, linearized state-feedback controller which penalises deviation from the expected trajectory under feedforward control. It is argued that the model captures natural features of human movement control, including implicit selection of reach time, and changes in movement variance dependent on the feedback condition. Other models typically specify movement time either arbitrarily or based on factors that do not result from the derivation of the control law.

Overall the paper is very well written, very clear and the modelling exercise appears to have been performed very cautiously. The paper complies with utmost scientific standards in terms of the accuracy of mathematical content, and clarity of the writing. However, I emphatically disagree with the relevance of the model to explain features of human sensorimotor control. The reason is that several modelling choices are simply wrong; the first being that reaching with or without vision is equated to open loop or closed loop control, which cannot be less compatible with the physiology. A second critical modelling choice is the use of a control model that consists in trajectory tracking, which is motivated for mathematical reasons to derive locally optimal feedback control laws around a nominal or expected trajectory, yet the possibility that the human brain uses trajectory tracking is debated, and evidence thereof is elusive if existing.

The first point is the fact that control without vision is not feedforward control. There is extended evidence that online feedback control uses limb afferent feedback to correct for error when the hand is unseen (Goodale, Pelisson and Prablanc, 1986, Nature, and following literature). Evidence for sophisticated feedback control based on limb afferent feedback has been extensively discussed and reviewed for instance in Scott, 2016 (TINS, 39 (8), 512-526). It is clear that vision contributes to reducing the variability of movement but the larger amount of variability observed when vision is absent should not be equated to feedforward control in a model of human reaching. Recent evidence of subtle effects of vision during reaching has been documented here (ito and Gomi, eLife, 9:e52380), and it is clear that changing visual information is not equivalent to open loop versus closed loop.

The second main contentious point is the hypothesis of trajectory tracking, which has been questioned in the work of Todorov and Jordan, 2002, in the task with different numbers of via-points. This led to partially artificial debate about whether there is or not a trajectory tracking mechanism in the brain. In fact, it appears that the brain may or may not use a trajectory for movement control, it is a feature of task related instructions (Cluff and Scott, JNS 35(36):12465–12476). These two points are major limitations: that the paper uses open loop trajectories to model movements without vision is in complete disagreement with current evidence that 1- such open loop trajectory may not exist in the brain, and 2-movements without vision are not feedforward.

It does not take anything away from the quality of the modelling work but as a model of human sensorimotor control, the model assumptions are not valid. I find it difficult to make a clear recommendation, the study could be published as is if the authors acknowledge that it is a bio-inspired model of control but it is not a model of the human sensorimotor system. Should the authors wish to make it a candidate model in the field of neurophysiology, then they should profoundly revise the underlying assumptions and abandon the idea of feed forward control in the absence of vision.

Major points:

The motivation for the model is not clearly apparent, it is interesting that a model can make predictions about movement time, but it is also not a big problem to consider it is a free parameter, that is we can decide movement time (i.e. I want to move slower or faster). Is there a genuine need for a model in which movement time is not under volitional control? Is there an experimental manipulation other than the empirical observation that movement time vary, which would justify that this parameter is linked to biomechanics and noise factors?

In general, the work by the groups of Shadmehr and Ahmed on the cost of time and temporal discounting of reward is acknowledged but under-represented. In fact, it is not clear that the explanation of movement time based on biomechanics only is required at all given that the temporal discounting of reward is a well-established phenomenon and that movement time is also at least partially under volitional control. Thus the ability to account for changes in movement time with biomechanical factors is likely valuable for specific conditions.

Specific points:

Line 70: Stochastic optimal control considers state-feedback adjustments not necessarily linked to corrections for changes relative to a plan

Equation 3: Why not using the formalism of stochastic differential equations? The proposed equation is slightly abusive as dw almost certainly does not exist (i.e. it has infinite instantaneous variation with probability 1).

Equation 9: It would be useful to unpack where the linearization comes from and show the derivation from the original system.

Reviewer #2: This study presents a new model for the planning and control of reaching movements. This model combines open-loop control with feedback control and is able to explain both the timing and the variability of reaching movements with and without sensory feedback. To explain the movement duration, a cost function is used that includes both effort and variance resulting from constant and signal-dependent noise in the motor command. The predictions are compared to previously published data and to the data of a new experiment.

General

Modelling the planning and control of reaching movements has a long history in the field of computational motor control. Over the years, many different ingredients have been proposed to be essential, such as open-loop control, feedback control, motor noise, and cost functions that include factors such as smoothness, effort and variance. This model can be considered as a unique combination of earlier proposed ingredients. The approach is therefore not original, but that’s not a problem. If this particular model has more predictive power than earlier models, this is scientific progress and an important result. To what extent that is the case for this study will be evaluated below.

The methodology seems generally sound and the paper is generally clearly written, although the writing can be improved as outlined below.

Specific

Major points

1. The model proposed here (SFFC) includes a feedforward and a feedback component. The feedforward-only version of the model is used to explain reaching without visual feedback, whereas the full version is used to explain reaching with visual feedback. It is however not correct to consider reaching without vision as reaching without any sensory feedback as there is always proprioceptive feedback. Reaching without any sensory feedback is only possible for deafferented patients. I therefore recommend to use data of such patients to test the predictions the feedforward-only version of the model, and to use the full model with increased sensory variance for reaching without visual feedback (but with proprioception).

2. The relevance of a model is determined by the quality of its predictions. The better a model predicts actual behaviour, the more relevant the model. The authors demonstrate that their model can explain many characteristics of human reaching movements. However, Figure 6 shows that there is one important aspect of reaching movements that the model does not predict correctly: the velocity profile. Whereas actual velocity profiles are well-known to be bell shaped, with the peak velocity occurring slightly before half the movement duration (see also the authors’ own data in Fig 6A,B), Figure 6C,D shows that the model predicts atypical velocity profiles with an initially slowly increasing speed, a peak velocity that is reached long after half the movement duration, followed by a rapid decrease to zero. Since these predicted velocity profiles differ strongly from actual velocity profiles, I cannot consider this as a model that predicts all important aspects of reach trajectories correctly. I recommend the authors to try to fix the model so that it predicts more realistic velocity profiles.

3. Equation 12 suggests that the model corrects for both task-relevant and task-irrelevant deviations from the nominal plan. Is that correct? This is at odds with stochastic optimal feedback control which corrects only for task-relevant deviations, a phenomenon that has strong experimental support. I recommend modifying the model so that it does not correct for task-irrelevant deviations.

4. The proposed model has a number of free parameters. The values of these parameters were determined on the basis of data of Wang et al. (2016), and then later used to make predictions for the new experiment. In principle, that’s a clean approach. However, in L254ff it is mentioned that some of the parameters were changed to compare the model predictions to the data. Are these then still model predictions, or is this just a fit of the model? I found this part of the paper rather vague. There is nothing wrong with fitting your new model, but why then first pretend as if the model was first fit to an independent data set, so that proper predictions can be made?

5. To simulate movement with visual feedback, it is assumed that the visual feedback provides information about the full state. However, the state includes joint angles, joint angles velocities and net joint torques (Eq 3). It’s debatable whether vision provides information about joint angles (I don’t look at my elbow during reaching), but it is certainly questionable that it provides information about joint angle velocities. Most problematic however, is the assumption that vision provides feedback about net joint torques - that’s certainly not realistic.

Minor points

The authors used ad hoc, non-standard acronyms for the various modelling approaches, such as DOC, SOC and SOOC. The logic of these acronyms is not always clear to me. For instance, SOC is used for stochastic optimal feedback control. Why not SOFC, as feedback is in my opinion the most crucial element of this approach?

L41-42: “SOC does not account for (…) the larger variance exhibited by movements without vision.” That’s not true, by increasing the level of variance in the sensory signals, this model predicts larger variance. 

L56: greater -> longer

The Introduction is very long (it ends only after 201 of the 475 lines of text). In my opinion, L28-76 are the genuine introduction, whereas L77-201 could be absorbed into the Methods section.

Fig 2, legend: I guess the corrective motor command can also be triggered by the consequences of noise in u.

L110-111: “the originality of our proposal is to assume that the primary goal of motor planning is to build a feedforward motor command.” That does not sound very original. All the models not including feedback have this same goal. 

Eq 3: what is omega_t?

L147-149: “Note that hard constraints for the final mean or covariance of the state can also be added in this formulation. We did so for the final mean state to ensure that the arm exactly reaches the desired target on average…” I don’t see why the final mean state had to be set. It’s more natural to use a cost function such as mean squared error that allows for a biases that are accompanied by lower variance. For instance, the well-known undershoot of saccades has been explained this way (Chris Harris), whereas it allowed Liu & Todorov (2007) to explain incomplete corrections for target jumps. 

L175-176: Please back up the claim that delays in the sensory feedback are not critical.

Methods: what time step was used in the model simulations, and do the model predictions depend on this time step?

L264: than -> as

L279ff: “whose position was displayed on a monitor placed in from of the participant”. Whose position was displayed? Placed in FRONT of the participant? How was the monitor oriented?  

L295: “and reappeared at the end”. When exactly?

L300: So there were only 5 repetitions of each movement? That’s way too few to obtain reliable estimates of the variability. At least several tens of repetitions are needed for this.

L307: what was the order of the filter?

L329: larger -> longer

L333: smaller -> shorter

L340: Fig 5C -> Fig 5D

L347-400: The results of a large number of statistical tests are reported here. It is however unclear to me why all these tests were conducted. A statistical test is conducted to test a hypothesis. However, none of the tests addresses a hypothesis raised in this study. Instead, the focus of the study is on the model and the extent to which it can explain observed reaching movements. The comparison between predictions and data therefore deserves much more attention than the statistical tests on the data, but very little effort is put in this comparison. In summary, I recommend to remove all the statistical tests on the data and replace these by a critical comparison between the data and model predictions.

L409-410: “This learning will minimize feedback…” Unclear what is meant.

L417-418: “One original aspect of SFFC is that the feedforward motor command considers uncertainty about the task dynamics…” This is not really original. Harris & Wolpert did this more than 20 years ago. 

L451: plan -> predict

Reviewer #3: In the current manuscript, the authors developed a new motor control model that explains the characteristics of timing (speed and duration) and variability of human reaching movements under the conditions of with and without visual feedback. The model which they call SFFC combines a feedforward control (SOOC in their model) and an online feedback control (SOC). The SOOC calculates an expected state trajectory and feedforward motor command in the planning phase and generates the movement in a feedforward manner. The critical point is that the motor command is optimized for the system stochasticity that includes not only signal-dependent noise, which is well known as a determinant of motor control, but also additive motor noise, which has not been given much importance in the field. As these two types of noise have opposing effects on the expected cost when the movement duration becomes longer, the optimal duration to minimize the cost is determined by the balance between the two. The motor planning by the SOOC can explain the timing characteristic that the movement duration increases as the target distance increases. Furthermore, it can account for an interesting observation reported by Wang et al (2016), which previous models fail to explain, that the duration of human preferred movements is greater than the minimum variance duration. On the other hand, the SOC corrects deviations online from the pre-planned state trajectory in a locally optimal manner. The SOC is assumed to be active only when visual feedback is available, which explains the variability characteristic that the variability is reduced in the presence of vision compared to when visual feedback is not available. The validity of their model was evaluated through comparison with human reaching data.

I think this model is original in that it explains the timing and variability characteristics without any ad hoc constraints, unlike previous models, but only by considering neuromechanical factors (noise properties, etc.). Also, this study tells us the importance of the motor planning considering additive motor noise in addition to signal-dependent noise. Thus, this study will bring new insight on the computational mechanisms underlying human motor control. However, I have three main concerns that will need to be addressed for this study to make a clear and reliable contribution to the literature.

1) First, the authors need more discussion on the currently dominating model, optimal feedback control model (OFC) (Todorov and Jordan, 2002). I agree that the OFC underestimates the importance of motor planning. However, the OFC is still a powerful model that can explain many features of human motor control. In order for the reader to better understand the difference in explanatory power between the OFC and the SFFC, the authors should more clearly mention (A) things the OFC can also explain with some additional assumptions, albeit ad hoc, and (B) things the OFC can explain but SFFC cannot if any.

(A): I acknowledge that the authors discuss carefully that the OFC can explain the timing characteristics by adding some assumptions as in previous studies. However, I feel the discussion on the difference between the variability with and without vision is insufficient. I wonder if the OFC could also explain an increased variability in the absence of vision when the weight of the observation noise increases due to only proprioceptive feedback available. If so, the authors should mention it.

(B): As the SOC corrects only task-relevant movements (minimal intervention principle), variability (task-irrelevant) in the middle of the movement is greater than at the endpoint, which is consistent with human experimental data (Todorov and Jordan, 2002). In contrast, the SFFC corrects any deviation from a planned trajectory at every moment. Thus, I wonder if the SFFC cound not explain the greater mid-movement variability. I would recommend that the authors add some discussion of this point.

2) The second is regarding the interpretation of the experimental data. While the data for variability (Fig 7A,) clearly supports the validity of the model, the data for timing (Fig 7C, E) does not appear to be consistent with the model simulation. On line 456, the authors state that “the fact that movements with and without vision had similar timing in our data reaffirmed the importance of ….”. However, the ANOVA analysis in fact provides marginal significances with p=0.055 (duration) and p=0.052 (peak velocity) for the main effect of the visual condition although the model predicts no difference. Thus, I ask the author to reconsider the interpretation of the data and discuss what possible causes make the difference.

3) The third is the contribution of proprioception in the absence of vision. The SFFC assumes that in the absence of vision only the SOOC is active. However, it seems more plausible to assume that the proprioceptive feedback (with a large observation noise) is available and the SOC still works. Although the authors state that the SOC with a large observation noise amounts to use SOOC (lines 454-455), I feel that is a bit extreme. If the SFFC assumes the SOC with a large observation noise is active in the absence of vision, can it explain the increased variability in the absence of vision?

Minor points

4) Figure1：The simulation result seems to change depending on the amount of additive noise and signal-dependent noise. When I run a simulation with Todorov’s model, I found the endpoint variance increases over the duration when the weight of additive noise is much greater than that of signal-dependent noise. The effect of increasing additive noise on endpoint variance is mentioned in Fig. 1B of Todorov (2005). Thus, the authors may modify the sentence in the caption, “The positional endpoint variance (gray trace) can also be seen to decrease and plateau to a value which mainly corresponds to that of visually-guided movements.”

5) Line 46: Please remove “A”.

6) Figure 4 AB: What do the black circles indicate? What are MD and VAR?

7) Figure 7: The presentation of experimental data requires a measure of variability.

**Have all data underlying the figures and results presented in the manuscript been provided?**

Reviewer #1: Yes

Reviewer #2: **No: **

Reviewer #3: Yes

PLOS authors have the option to publish the peer review history of their article (what does this mean?). If published, this will include your full peer review and any attached files.

Reviewer #1: No

Reviewer #2: No

Reviewer #3: No
---

## [Decision Letter · Decision Letter 1]

24 Mar 2021

Dear Dr. Berret,

Thank you very much for submitting your manuscript "Stochastic optimal feedforward-feedback control determines timing and variability of arm movements with or without vision" for consideration at PLOS Computational Biology.

As with all papers reviewed by the journal, your manuscript was reviewed by members of the editorial board and by several independent reviewers. In light of the reviews (below this email), we would like to invite the resubmission of a significantly-revised version that takes into account the reviewers' comments.

---------------

As you will see below review 1 (and indirectly the response to the previous version by reviewer 2) indicate that the equating of movement without vision and feedforward-only control is erroneous, and that the claim is still present in the manuscript.

This needs to be clear in the paper that movement without vision can not be fully explained by feedforward mechanisms. This may require a minor or major rewrite, depending on whether you agree with this statement.

Of course it might be possible instead to test experimentally (using deafferented patients, or with perturbation study) whether the feedforward aspect of the model alone can explain explain movement without vision, but that may be going further than your aim with this paper.

----------------

We cannot make any decision about publication until we have seen the revised manuscript and your response to the reviewers' comments. Your revised manuscript is also likely to be sent to reviewers for further evaluation.

Sincerely,

Ulrik R. Beierholm

Associate Editor

PLOS Computational Biology

Samuel Gershman

Deputy Editor

PLOS Computational Biology

Reviewer's Responses to Questions

**Comments to the Authors:**

Reviewer #1: The authors have extensively revised the manuscript but I admit I feel that the idea of feedforward control corresponding to movements performed without vision is still there. Movements performed without vision are now associated with larger uncertainty but this modification falls a bit short of the more profound change that I believe was necessary.

To take a few examples, the abstract states the following: “[feedback] to correct task-related errors which are mostly noticeable when vision is available”. This is wrong and misleading, there is extended evidence for non-visually mediated corrections for task-related errors. In the authors’ summary, it is written “feedforward motor command before the movement starts, which can be observed for example in arm movements without vision […]”. I do not feel it is useful to reiterate that movements without vision are not feedforward. This idea is pushed forward again later in the intro where it is written that: “SOC is designed to account for visually guided movements…” is also wrong. There is a double fallacy in this statement, first it ignores that feedback in SOC models needs not be visual, second it implies that when there is no vision SOC is probably not useful. This pushes the idea that movements without vision are feedforward, which I indicated in my first review does not stand one second in front of evidence that movement without vision are controlled in closed loop. The overall experimental work compares feedforward with movements performed without vision. The statement in the discussion that movement are largely based on feedforward control is again not supported.

The main message put forward is that changes in movement time can be captured by considering open loop minimization in the brain. This is interesting, but that does not imply that a feedforward controller is derived, much less executed. The experimental validation is not adequate and therefore the model is not supported. It is also conceivable that movement times are selected based on prior experience (including optimisation criteria, this is a contribution of the authors’), and that the implicit selection of movement time is used in feedback controllers with or without vision. To argue that movement performed without vision are similar to feedforward and establish that the selection of movement time could be due to the optimization of a feedforward controller requires at least a task that involves perturbations. In support to their model the authors must demonstrate in a perturbation task that there is less or no feedback correction when there is no vision.

Much of the authors’ reasoning to establish a similarity between no-vision and feedforward control is based on movement variability, but this reasoning is inconsistent with the work of Ito and Gomi (2020, eLife). The authors argumentation imply that corrections are reduced when vision is withheld, but there is another phenomenon, that is the gain of long-latency responses in this case in fact increased. Long-latency responses are not impedance control, and responses in the same time window increase without vision. This produced an increase in variance, but not due to less feedback correction, instead it was due to more vigorous corrections. This is a sufficient piece of evidence to reject the authors’ model. That the model produces movement times that are consistent with human behaviour is interesting but the analogy stops there.

Other major points:

1. The limitation of SOC relative to changes in movement time is a straw-man, it simply follows from finite horizon formulation, so this model is not supposed to produce changes in movement time. The argumentation must be reworked throughout.

2. Aspects of model presentation must be clarified: the procedure considering “fixed or free movement time” (just prior to Eqn. 4) was unclear and should be sketched out; pls sketch the argument why P(t) appears in the deterministic part of the SDE; it is unclear what manipulation was performed with the hard constraints that make the terminal penalty cost redundant or useless, what was the manipulation exactly?

3. Page 7 middle: pls clarify, it is suggested that deafferented patients have both similar and higher level of variability, this section was unclear. The argument that variability in deafferented patients is comparable in some conditions to control groups is not telling that they use the same control strategy, again a perturbation paradigm would clearly highlight differences between the two.

4. The correlation between timings of movements performed with or without vision is taken as evidence that they can share the same feedforward control command, an equally valid conclusion is that the consistent timing originated from the same state-feedback controller. This does not undermine the authors’ efforts to show that movement times may be selected optimally, although there is no conclusive evidene that this selection is based on a feedforward controller.

5. While the model parameters have likely been selected carefully it is necessary to perform a sensitivity analysis to assess the robustness of the model simulations against variations in parameter settings.

Reviewer #3: The new version submitted by the authors contains significant updates in their model as well as additional explanation/discussion on the model simulation and experimental data to answer to the points we raised in our first review. I appreciate the additional effort and think that the authors improved the quality of the paper which I consider now suitable for publication.

**Have all data underlying the figures and results presented in the manuscript been provided?**

Reviewer #1: Yes

Reviewer #3: Yes

PLOS authors have the option to publish the peer review history of their article (what does this mean?). If published, this will include your full peer review and any attached files.

Reviewer #1: No

Reviewer #3: No
---

## [Decision Letter · Decision Letter 2]

5 May 2021

Dear Dr. Berret,

We are pleased to inform you that your manuscript 'Stochastic optimal feedforward-feedback control determines timing and variability of arm movements with or without vision' has been provisionally accepted for publication in PLOS Computational Biology.

Best regards,

Ulrik R. Beierholm

Associate Editor

PLOS Computational Biology

Samuel Gershman

Deputy Editor

PLOS Computational Biology

Reviewer's Responses to Questions

**Comments to the Authors:**

Reviewer #1: The authors made commendable efforts to take all previous concerns into account and I recommend that the manuscript be accepted for publication,

**Have the authors made all data and (if applicable) computational code underlying the findings in their manuscript fully available?**

Reviewer #1: Yes

PLOS authors have the option to publish the peer review history of their article (what does this mean?). If published, this will include your full peer review and any attached files.

Reviewer #1: No

---

## [Editor Report · Acceptance letter]

8 Jun 2021

PCOMPBIOL-D-20-02130R2 

Stochastic optimal feedforward-feedback control determines timing and variability of arm movements with or without vision

Dear Dr Berret,

I am pleased to inform you that your manuscript has been formally accepted for publication in PLOS Computational Biology. Your manuscript is now with our production department and you will be notified of the publication date in due course.

With kind regards,

Katalin Szabo
